# The carboxyl-terminal sequence of PUMA binds to both anti-apoptotic proteins and membranes

**James M Pemberton**[1,2], **Dang Nguyen**[1,2], **Elizabeth J Osterlund**[2,3], **Wiebke Schormann**[2], **Justin P Pogmore**[2,3], **Nehad Hirmiz**[2,4], **Brian Leber**[5], **David W Andrews**[1,2,3]*

[1]Department of Medical Biophysics, Faculty of Medicine, University of Toronto, Toronto, Canada; [2]Biological Sciences Platform, Sunnybrook Research Institute, Toronto, Canada; [3]Department of Biochemistry, Faculty of Medicine, University of Toronto, Toronto, Canada; [4]Department of Biomedical Engineering, McMaster University, Hamilton, Canada; [5]Department of Medicine, McMaster University, Hamilton, Canada

*For correspondence: david.andrews@sunnybrook.ca

**Competing interest:** The authors declare that no competing interests exist.

**Abstract** Anti-apoptotic proteins such as BCL-X$_L$ promote cell survival by sequestering pro-apoptotic BCL-2 family members, an activity that frequently contributes to tumorigenesis. Thus, the development of small-molecule inhibitors for anti-apoptotic proteins, termed BH3-mimetics, is revolutionizing how we treat cancer. BH3 mimetics kill cells by displacing sequestered pro-apoptotic proteins to initiate tumor-cell death. Recent evidence has demonstrated that in live cells the BH3-only proteins PUMA and BIM resist displacement by BH3-mimetics, while others like tBID do not. Analysis of the molecular mechanism by which PUMA resists BH3-mimetic mediated displacement from full-length anti-apoptotic proteins (BCL-X$_L$, BCL-2, BCL-W, and MCL-1) reveals that both the BH3-motif and a novel binding site within the carboxyl-terminal sequence (CTS) of PUMA contribute to binding. Together these sequences bind to anti-apoptotic proteins, which effectively 'double-bolt locks' the proteins to resist BH3-mimetic displacement. The pro-apoptotic protein BIM has also been shown to double-bolt lock to anti-apoptotic proteins however, the novel binding sequence in PUMA is unrelated to that in the CTS of BIM and functions independent of PUMA binding to membranes. Moreover, contrary to previous reports, we find that when exogenously expressed, the CTS of PUMA directs the protein primarily to the endoplasmic reticulum (ER) rather than mitochondria and that residues I175 and P180 within the CTS are required for both ER localization and BH3-mimetic resistance. Understanding how PUMA resists BH3-mimetic displacement will be useful in designing more efficacious small-molecule inhibitors of anti-apoptotic BCL-2 proteins.

## Editor's evaluation

This is an important study investigating interactions of the pro-apoptotic PUMA with anti-apoptotic BCL-2 proteins. The authors demonstrate convincingly that the PUMA/BCL-2 interactions are mediated not only via BH3-domain interaction, but also depend on a C-terminal sequence of PUMA similar to BIM. Unexpectedly they find PUMA is often localising to the ER. This manuscript is important for researchers focusing on cell death and anti-tumor drug development.

## Introduction

Apoptosis, a form of programmed cell death, is an essential physiological process responsible for the elimination and disposal of malignant or excessive cells in multicellular organisms (*Fuchs and Steller, 2011*). The loss of outer-mitochondrial membrane integrity, known as mitochondrial outer-membrane

(MOM) permeabilization (MOMP), is regarded as an irreversible event, resulting in the release of cytochrome c and apoptotic factors from the mitochondria that leads to the activation of caspases and cell death (*Kale et al., 2018*). MOMP is tightly regulated by BCL-2 protein family proteins, including both pro- and anti-apoptotic regulators that share from one to four homology motifs (BH1 to BH4). BCL-2 and its anti-apoptotic homologs possess all four BH motifs and keep the effector pore-forming multi-BH domain pro-apoptotic proteins, BAX and BAK, from inducing MOMP through direct binary protein-protein interactions (*Kale et al., 2018*; *Bogner et al., 2020*). Anti-apoptotic proteins can also prevent MOMP by binding BH3-only proteins, which are pro-apoptotic proteins that share only the BH3 region with other BCL-2 family proteins. The interaction of pro-apoptotic proteins with anti-apoptotic proteins, which is often but not always binary, results in a mutual sequestration that inhibits both proteins. BH3-only 'activator' proteins, like BIM and BID, bind to and activate BAX and BAK. BH3-only 'sensitizer' proteins, like BAD and HRK, bind to and inhibit anti-apoptotic proteins but do not activate BAX or BAK. Unlike the other BH3-proteins where there is a clear distinction, whether the BH3-protein PUMA functions primarily as an inhibitor of anti-apoptotic proteins (sensitizer) or as an activator of BAX and BAK remains unclear and may depend on the cell type analyzed. Although first identified as a p53 transcriptional target (*Yu et al., 2001*; *Nakano and Vousden, 2001*), PUMA has been characterized as a mediator of cell death induced by DNA damage, endoplasmic reticulum (ER) stress and oxidative damage. All these stresses are induced by common chemotherapeutics and ionizing radiation (*Jeffers et al., 2003*; *Jiang et al., 2006*) suggesting that PUMA may play a role in chemotherapy responses. As the overexpression of pro-survival BCL-2 proteins is not only a hallmark of cancer progression but also critical for tumor cells to sustain a high proliferative rate and survive genomic instability, apoptosis modulation has been proposed as a therapeutic approach to selectively target cancer cells for elimination (*Hanahan and Weinberg, 2011*; *Delbridge et al., 2016*).

The development of small molecule inhibitors that mimic the BH3-motif of pro-apoptotic BCL-2 family proteins and function as competitive inhibitors for BH3-protein binding to anti-apoptotic proteins (BH3-mimetics) is an area of active academic and pharmaceutical research. BH3-mimetics result in the death of cancer cells that depend on anti-apoptotic proteins for survival (*Delbridge et al., 2016*). Among BH3-mimetics, the selective BCL-2 inhibitor, Venetoclax (ABT-199) is leading advancement to the clinic, and is FDA approved for the treatment of relapsed chronic lymphocytic leukemia and acute myeloid leukemia in patients not fit for standard induction chemotherapy (*Delbridge et al., 2016*; *Roberts et al., 2016*). Navitoclax (ABT-263), a BH3-mimetic that inhibits BCL-2, BCL-X$_L$, and BCL-W results in undesirable but manageable on-target thrombocytopenia due to the dependence of platelets on BCL-X$_L$ (*Tse et al., 2008*). Unexpectedly complexes of anti-apoptotic proteins bound to PUMA and BIM, are highly resistant to all known BH3-mimetics (*Aranovich et al., 2012*; *Liu et al., 2019*). Resistance can be partly attributed to membrane binding by both the BH3-protein and its anti-apoptotic target increasing the local concentrations and therefore binding interactions (*Liu et al., 2019*; *Pécot et al., 2016*). However, we recently demonstrated that in addition to the BH3-sequence, there is a second anti-apoptotic protein binding motif in the carboxyl-terminal sequence (CTS) of BIM (*Liu et al., 2019*). Together the two binding sequences enable BIM to 'double-bolt lock' to anti-apoptotic proteins. The increased avidity that results from two independent binding sequences is presumed to confer resistance to displacement by BH3 mimetics.

Similar to BIM, PUMA is an intrinsically disordered protein containing a BH3 motif important for binding other BCL-2 family proteins and a CTS reported to function as a tail-anchor that integrates the protein in the MOM (*Rogers et al., 2014*; *Wilfling et al., 2012*). However, unlike conventional tail-anchor sequences and the CTS of BIM, the CTS of PUMA contains multiple prolines and charged residues, and an unusually short span of hydrophobic amino acids (*Borgese et al., 2003*; *Mehlhorn et al., 2021*). Therefore, we investigated whether the unusual CTS of PUMA contributes to resistance to BH3-mimetics. Our results suggest that similar to BIM, PUMA 'double-bolt locks' the anti-apoptotic proteins BCL-X$_L$, BCL-2 and BCL-W. Unexpectedly, in multiple cell types, exogenously expressed PUMA is primarily localized at the ER and not the mitochondrial outer membrane (MOM). Furthermore, replacing the PUMA CTS with ER-specific tail-anchor sequences from other proteins resulted in PUMA mutants that bound specifically to ER membranes, retained pro-apoptotic function as a sensitizer binding to anti-apoptotic protein(s) but were not resistant to BH3-mimetics, suggesting that the sequence of the PUMA CTS rather than membrane binding is responsible for BH3-mimetic resistance. Mutagenesis and live cell experiments identified PUMA CTS residues I175 and P180 as required for

both ER localization and resistance to BH3-mimetics. Our data indicate that inhibition of both BH3 and CTS binding to anti-apoptotic proteins may be required to mobilize PUMA for the treatment of cancer.

## Results
### The PUMA CTS contributes to BH3-mimetic resistance independent of membranes in vitro

To understand how PUMA resists BH3-mimetic displacement from BCL-$X_L$, full-length recombinant BCL-$X_L$ (CAA80661) and full-length PUMA (AAB51243) were purified, and their interaction was measured in vitro using Förster resonance energy transfer (FRET) (*Pogmore et al., 2016*). Previously, it was shown that PUMA and BCL-$X_L$ complexes are highly resistant to BH3-mimetic displacement when bound to membranes (*Pécot et al., 2016*), and that mutations in the BCL-$X_L$ CTS abolished both membrane binding and BH3-mimetic resistance (*Pécot et al., 2016*). However, PUMA also contains a CTS that binds the protein to membranes (*Wilfling et al., 2012*; *Yee and Vousden, 2008*) and it remains unclear whether it contributes to BH3-mimetic resistance (*Liu et al., 2019*). Therefore, PUMA with a deletion of the last 26 amino acids (PUMA-d26) that encompasses the CTS was also purified. To measure function, purified PUMA protein was added to a 'SMAC-mCherry MOMP assay', as previously described (*Chi et al., 2020*). In brief, mitochondria were isolated from baby mouse kidney (BMK) cells deficient for *Bax* and *Bak* (BMK-dko cells) expressing a fluorescent protein (FP), mCherry, fused to the N-terminal sequence of SMAC (SMAC-mCherry). SMAC-mCherry localizes to the intermembrane space of mitochondria, and is released upon MOMP. Isolated SMAC-mCherry containing mitochondria were reconstituted with recombinant BCL-2 family protein(s), and % SMAC-mCherry release (MOMP) was measured by monitoring SMAC-mCherry fluorescence in the supernatant after pelleting mitochondria, relative to the detergent-induced full permeabilization.

As expected, the addition of each BH3-protein alone was insufficient to induce MOMP as indicated by the low % Smac-mCherry Release (≈20%) (*Figure 1A*, columns 2,3,9,10). However, the combination of recombinant tBID [4 nM] and BAX [20 nM] was sufficient to induce MOMP at ≈60% release (column 4), which was inhibited by [10 nM] of BCL-$X_L$ (column 5). Addition of the BCL-$X_L$ inhibitor BAD [50 nM] (column 6), recombinant PUMA (column 7) or PUMA-d26 (column 8) was sufficient to functionally inhibit BCL-$X_L$, thereby inducing SMAC-mCherry release (MOMP) in a BAX-dependent manner (*Figure 1A*). This confirms that recombinant PUMA and PUMA-d26 function as sensitizers by inhibiting BCL-$X_L$ to indirectly activate BAX.

We next tested PUMA pro-apoptotic function in live cells by transient transfection of HEK293, Baby mouse kidney (BMK) and HCT116 cell lines and assessed cell death in the transfected cells by Annexin V staining (*Figure 1—figure supplement 1A*). Here we use a Venus-fused to the N-terminus of the BH3-only protein BIM ($^V$Bim) or PUMA ($^V$PUMA) to identify the transfected cells expressing the BH3-proteins by fluorescence intensity. As we previously published (*Chi et al., 2020*), HEK293 cell lines are unprimed and can only be killed efficiently by an activator protein such as BIM (60% Annexin V positive), whereas PUMA and the previously established sensitizer-only BIM mutant (BIM-dCTS) did not induce cell death in HEK293. On the other hand, BIM, PUMA and PUMA-d26 killed BMK and HCT116, which are primed cells that can be killed by a sensitizer protein (*Chi et al., 2020*). Similar to sensitizer proteins, BH3-mimetics are designed to target anti-apoptotic proteins and do not activate BAX or BAK. Thus, we used a combination of two BH3-mimetics S63845 (MCL-1 inhibitor) and ABT-263 (BCL-$X_L$, BCL-2, and BCL-W inhibitor) to validate the apoptotic priming of these stable cell lines by doing a two-way titration of the two drugs (*Figure 1—figure supplement 1B*). Upon BH3-mimetics treatment, activator proteins and/or active BAX/BAK are displaced from the antiapoptotic proteins in the primed cells resulting in cell death. The combination of these drugs at 10 μM resulted in 47% and 73% Annexin V positivity in HCT116 and BMK, respectively, but did not increase Annexin V positivity in HEK293, confirming that HEK293 is the only unprimed cell line among the three. Taken together, the data show that PUMA acts as a sensitizer and that its CTS is not required for sensitizer function, as truncated PUMA without its CTS can still bind to and inhibit BCL-$X_L$.

Binding of recombinant Alexafluor 647 ($A^{*647}$) labeled BCL-$X_L$ to Alexafluor 568 ($A^{*568}$) PUMA and PUMA-d26 was measured directly by FRET in the presence of liposomes with a phospholipid composition similar to that of mitochondria (*Figure 1B–E*). In the absence of BH3-mimetics, (indicated as

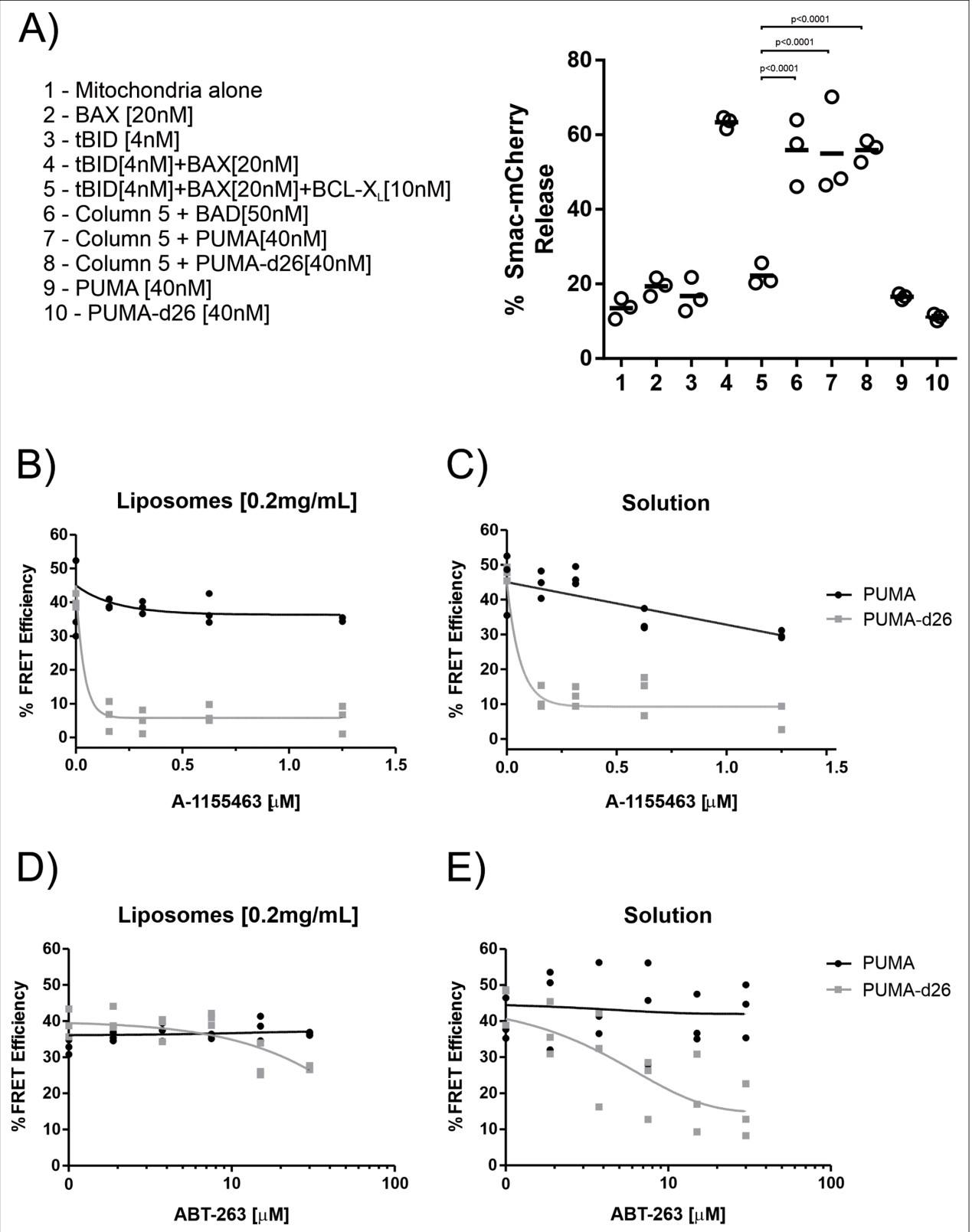

**Figure 1.** The PUMA CTS contributes to BH3-mimetic resistance independent of membranes in vitro. (**A**) Left panel, legend indicating the combinations of recombinant proteins incubated with purified mitochondria encapsulating SMAC-mCherry. Right panel, release of SMAC-mCherry from mitochondria by the combination of recombinant proteins indicated by the legend numbers below. Each point represents a single technical replicate. The horizontal bar indicates the average of all three technical replicates per group. A one-way ANOVA and Dunnett's multiple comparisons test resulted in the

*Figure 1 continued on next page*

*Figure 1 continued*

indicated p-values. (**B–E**) Alexa*568 labelled PUMA [5 nM] (black line) or PUMA-d26 [5 nM] (grey line) were incubated with Alexa*647 labelled BCL-X$_L$ [40 nM] in the presence of the indicated concentration of the BH3-mimetics (**B and C**) A-1155463 or (**D and E**) ABT-263. Each graph includes datapoints from three independent replicates. Total data were fit to a one phase exponential decay (grey and black lines as indicated). (**B,D**) Incubations of contained mitochondrial-like liposomes (0.2 mg/mL). (**C,E**) Solution indicates incubations that did not contain liposomes.

The online version of this article includes the following figure supplement(s) for figure 1:

**Figure supplement 1.** PUMA functions as a sensitizer in cells and kills BMK and HCT116 cells but not HEK293 cells.

Concentration, 0 nM) the FRET efficiency was ~40% between the donor PUMA$^{*A568}$ and the acceptor BCL-X$_L$$^{*A647}$. The addition of the selective BCL-X$_L$ BH3-mimetic, A-1155463, did not result in a significant decrease in FRET efficiency, demonstrating that binding of PUMA$^{*A568}$ to BCL-X$_L$$^{*A647}$ remains unchanged (*Figure 1B*, black line). A FRET efficiency of ~40% was also measured with the same concentrations of PUMA-d26$^{*A568}$ incubated with BCL-X$_L$$^{*A647}$, indicating similar protein binding (*Figure 1B*, grey line). However, the addition of low concentrations of A-1155463 reduced FRET efficiency to less than 10%, demonstrating BH3-mimetic mediated displacement of PUMA-d26$^{*A568}$ from BCL-X$_L$$^{*A647}$ (*Figure 1B*). Thus, removing the PUMA CTS increases PUMA susceptibility to BH3-mimetic displacement from BCL-X$_L$.

When the same experiment was performed in solution, full length PUMA$^{*A568}$ was more resistant to BH3-mimetic displacement than PUMA-d26$^{*A568}$ suggesting that the CTS of PUMA contributes to BH3-mimetic resistance by binding to BCL-X$_L$ even in the absence of membranes (*Figure 1C*). To corroborate the result the experiment was repeated using the less potent but better studied BCL-X$_L$ inhibitor ABT-263. In the presence of liposomes, both PUMA$^{*A568}$ and PUMA-d26$^{*A568}$ resisted displacement from BCL-X$_L$$^{*A647}$ by ABT-263 (*Tao et al., 2014*; *Figure 1D*) but in solution, PUMA-d26$^{*A568}$ was displaced from BCL-X$_L$$^{*A647}$ (*Figure 1E*), consistent with the CTS of PUMA contributing to BH3-mimetic resistance by binding to BCL-X$_L$ independent of binding PUMA to membranes.

## The PUMA CTS contributes to BH3-mimetic resistance in live cells

To determine if our findings with purified proteins replicate what occurs in live cells, we used quantitative fast fluorescence lifetime imaging microscopy – Förster resonance energy transfer (qF$^3$) to measure PUMA binding to anti-apoptotic proteins (*Osterlund et al., 2022*). For these measurements, the donor fluorescent protein mCerulean3 was stably expressed as a fusion to the N-terminus of the indicated anti-apoptotic protein in BMK-dko cells in which the *Bax* and *Bak* genes are deleted. The acceptor protein of the FRET pair - Venus (V) fused to the N-terminus of PUMA ($^V$PUMA) was expressed in the cells by transient transfection. Four hours later the media was exchanged to add the indicated BH3-mimetic or DMSO as a solvent control. Twenty hr later, the cells were analyzed by qF$^3$. To assess the effect of mutations in PUMA on binding to the anti-apoptotic proteins, BMK-dko cell lines expressing either $^C$BCL-X$_L$, $^C$BCL-2, or $^C$BCL-W were transfected with plasmids encoding variants of $^V$PUMA as previously described (*Pemberton et al., 2019*). In these assays, binding of both $^V$PUMA and $^V$PUMA-d26 to $^C$BCL-X$_L$, $^C$BCL-2 and $^C$BCL-W was established with approximately equal apparent dissociation constants (6–9 μM) by ensuring that in the cell images analyzed the expression level of the donor exceeded the absolute dissociation constant (*Figure 2A*, DMSO lanes) (*Osterlund et al., 2022*). In these experiments, the non-binding mutants with a 4E mutation within the BH3 motif (BH3-4E) of $^V$PUMA and $^V$PUMA-d26 were used to control for collisions as opposed to binding interactions (*Figure 2A*, bottom panel of heatmap).

Regions of interest (ROIs) in the qF$^3$ images were identified automatically and fluorescence lifetimes were calculated as previously (*Osterlund et al., 2022*). The decrease in the 3.8ns fluorescence lifetime in $^C$BCL-X$_L$ expressing cells (blue) to less than 3.4 ns in cells expressing $^V$PUMA is indicative of a binding interaction (*Figure 2B*, red cells). However, to clearly differentiate binding from collisions that can occur at high frequency for colocalized proteins it is necessary to demonstrate that the qF$^3$ data can be better fit to Hill equation than to a straight line. For this purpose, values that are directly related to the fraction of donor molecules in the bound state, the fractional change in angular frequency (Δω) obtained from polar plots of the data, were plotted as a function of the concentrations of the unbound donor (Free Venus) and fit to a Hill equation to enable calculating apparent dissociation constants for each binding curve (*Osterlund et al., 2022*).

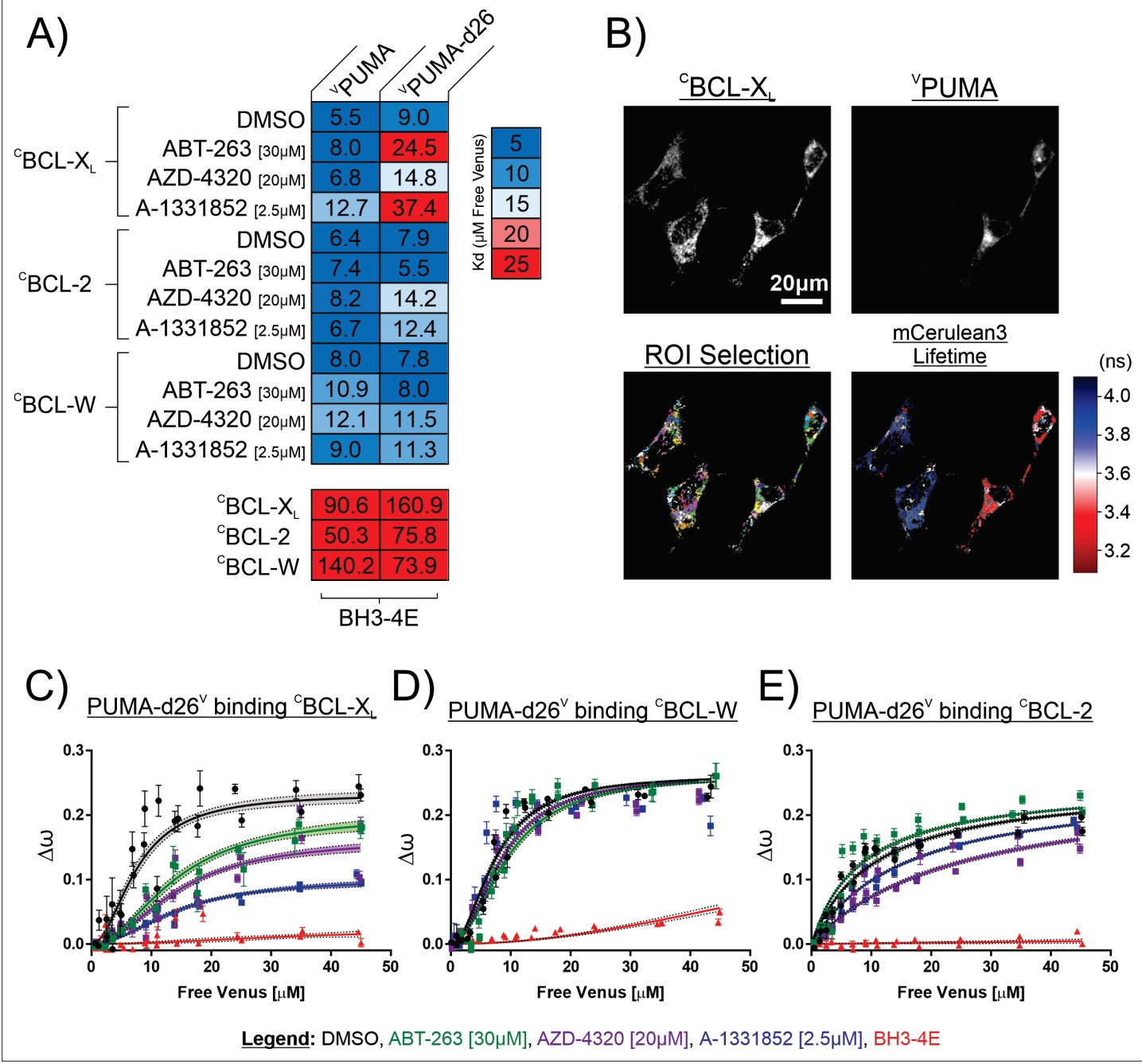

**Figure 2.** The PUMA CTS contributes to resistance to BH3-mimetics in live cells. Quantitative Fast FLIM-FRET (qF[3]) was used to measure binding of PUMA and PUMA-d26 to the anti-apoptotic proteins: CBCL-XL, CBCL-W, and CBCL-2 in live BMK-dko cells. (**A**) Calculated apparent dissociation constants (Kd's) for VPUMA and VPUMA-d26 binding to the indicated anti-apoptotic proteins are presented in a heatmap according to the scale at the right, with the calculated values specified inside the heatmap cells. VPUMA-d26 was displaced from CBCL-XL by all of the mimetics, as indicated by the increased apparent Kd in response to addition of BH3-mimetic. The protein pairs are indicated to the left and at the top. Final drug concentrations added to cells are indicated at the left. DMSO is the solvent control. BH3-4E below the panels indicates mutation of the BH3 protein listed at the top. Data is averages of three independent biological replicates. (**B**) Representative qF[3] micrographs of BMK-dko cells stably expressing CBCL-XL, and transiently transfected with the plasmid to express VPUMA. Regions of interest identified automatically (ROI Selection) were assigned arbitrary colors to permit visualization. The FLIM image indicates the subcellular localization of VPUMA- CBCL-XL protein complexes (red, decreased mCerulean3 fluorescence lifetime) compared to unbound CBCL-XL (blue). (**C–E**) The effect of BH3 mimetics on the binding of PUMA-d26V to the anti-apoptotic proteins indicated above the graphs as measured by qF[3]. Binding data (Δω,from phasor plots)at different concentrations of unbound PUMA-d26V (means, symbols; error bars, SE) was fit to a Hill equation. Data points are averages from independent experiments. Line was fit to the data points from all three independent experiments. Lines with shaded areas indicate 90% confidence interval for the best fit. The results demonstrate displacement in (**C**) from BCL-XL, but not

*Figure 2 continued on next page*

*Figure 2 continued*

in (**D**) from BCL-W and to an intermediate extent in (**E**) from BCL-2 when incubated with the drugs indicated by the legend. DMSO is the solvent control and BH3-4E indicates the non-binding PUMA-d26$^V$ mutant used to control for collisions.

The online version of this article includes the following figure supplement(s) for figure 2:

**Figure supplement 1.** $^V$PUMA binding to anti-apoptotic proteins is resistant to BH3-mimetics displacement in live cells due to the last 26 amino acids in the C-terminus of PUMA.

**Figure supplement 2.** Venus fused to the C-terminus of PUMA (PUMA$^V$, grey line) resulted in higher $\Delta\omega$ for binding to $^C$BCL-X$_L$ than when Venus was fused to the N-terminal of PUMA ($^V$PUMA, black line).

Addition of the dual BCL-2/BCL-X$_L$ inhibitors ABT-263 [30 µM], AZD-4320 [20 µM] or a newer BCL-X$_L$ inhibitor related to A-1155463, A-1331852 [2.5 µM] caused increases in the apparent dissociation constants that were more pronounced for $^V$PUMA-d26 than for $^V$PUMA binding to $^C$BCL-X$_L$ (***Figure 2A***). This result demonstrates that these BH3-mimetics displaced truncated PUMA (PUMA-d26), to a much greater extent than full-length PUMA. The most dramatic change in apparent K$_d$ was for A-1331852 which severely reduced the affinity for $^V$PUMA-d26 binding to $^C$BCL-X$_L$ resulting in an apparent K$_d$ increase from ~9 to~37 µM. In contrast, measurements of the effect of the same inhibitor on binding of full-length $^V$PUMA to $^C$BCL-X$_L$ revealed much less but detectable displacement (the apparent K$_d$ increased from ~6 to~13 µM) (***Figure 2A***). Consistent with these results, in live cells, A-1331852 is more selective and more potent than either AZD-4320 or ABT-263 (***Osterlund et al., 2022***).

The inhibitors ABT-263 and AZD-4320 are BH3-mimetics reported to target BCL-X$_L$, BCL-2 and BCL-W (***Tse et al., 2008***; ***Leverson et al., 2015***) and BCL-X$_L$ and BCL-2, (***Leverson et al., 2015***) respectively. When these drugs were used to probe PUMA binding to BCL-2, surprisingly ABT-263 had no effect and AZD-4320 only partially reduced the binding of $^V$PUMA-d26 with $^C$BCL-2, indicating that deletion of the PUMA CTS alone is not sufficient to make PUMA susceptible to displacement from BCL-2 (***Figure 2—figure supplement 1***). None of the inhibitors reduced binding of $^V$PUMA-d26 to BCL-W (***Figure 2—figure supplement 1***), however that might be due to poor binding of the inhibitors to the anti-apoptotic protein rather than to the binding affinity of PUMA for BCL-W (***Osterlund et al., 2022***).

Unexpectedly, fusing Venus to the C-terminus of PUMA (PUMA$^V$) resulted in a higher FRET efficiency with anti-apoptotic proteins (indicated by the increase in $\Delta\omega$), likely due to increased proximity between the donor and acceptor fluorescence proteins in the complex and inconsistent with the PUMA CTS spanning a membrane (***Figure 2—figure supplement 2***). A larger change in $\Delta\omega$ results in a greater dynamic range in the assay and therefore the possibility of detecting smaller changes in binding. Both PUMA$^V$ and PUMA-d26$^V$ bound to anti-apoptotic proteins in a BH3-dependent manner (***Figure 2—figure supplement 2***). Similar to the results obtained above, when binding to $^C$BCL-X$_L$ was measured, PUMA$^V$ resisted BH3-mimetic displacement, while PUMA-d26$^V$ was displaced (***Figure 2C-E***, ***Figure 3***). Indeed, binding was reduced to the point where the change in $\Delta\omega$ did not return to the level seen for the bound state even at 45 µM free PUMA$^V$ (***Figure 2C***, blue line). Similar to the results with Venus fused to the amino terminus, none of the mimetics displaced PUMA-d26$^V$ from BCL-W (***Figure 2D***) consistent with relatively poor binding of the mimetics to BCL-W. However, for BCL-2 visual inspection of the binding curves revealed that the mimetics, particularly AZD-4320, partially displaced PUMA-d26$^V$ (***Figure 2E***). Taken together our data indicate that the PUMA CTS contributes to PUMA binding to BCL-X$_L$ and BCL-2 in live cells.

## In cells PUMA resists BH3-mimetic displacement by double-bolt locking to BCL-2 family anti-apoptotic proteins

The resistance of PUMA to displacement from anti-apoptotic proteins by BH3-mimetics is similar to recently published results for the BH3-protein BIM (***Aranovich et al., 2012***; ***Liu et al., 2019***; ***Pécot et al., 2016***) yet the CTS of PUMA is very different than the CTS of BIM. To determine whether PUMA is also 'double-bolt locked' to anti-apoptotic proteins by the BH3-motif and CTS we constructed new mutants of PUMA. These mutants have the BH3-motif mutated (Lock 1), the CTS deleted (Lock 2), or both (Lock 1 and 2) mutated (***Figure 3A***). To disable the PUMA BH3 region sufficiently to measure the importance of the CTS separately but without abolishing the interaction entirely we replaced PUMA (residues 133–151) with the BH3 region from the protein BID. Unlike PUMA, BID is a pro-apoptotic

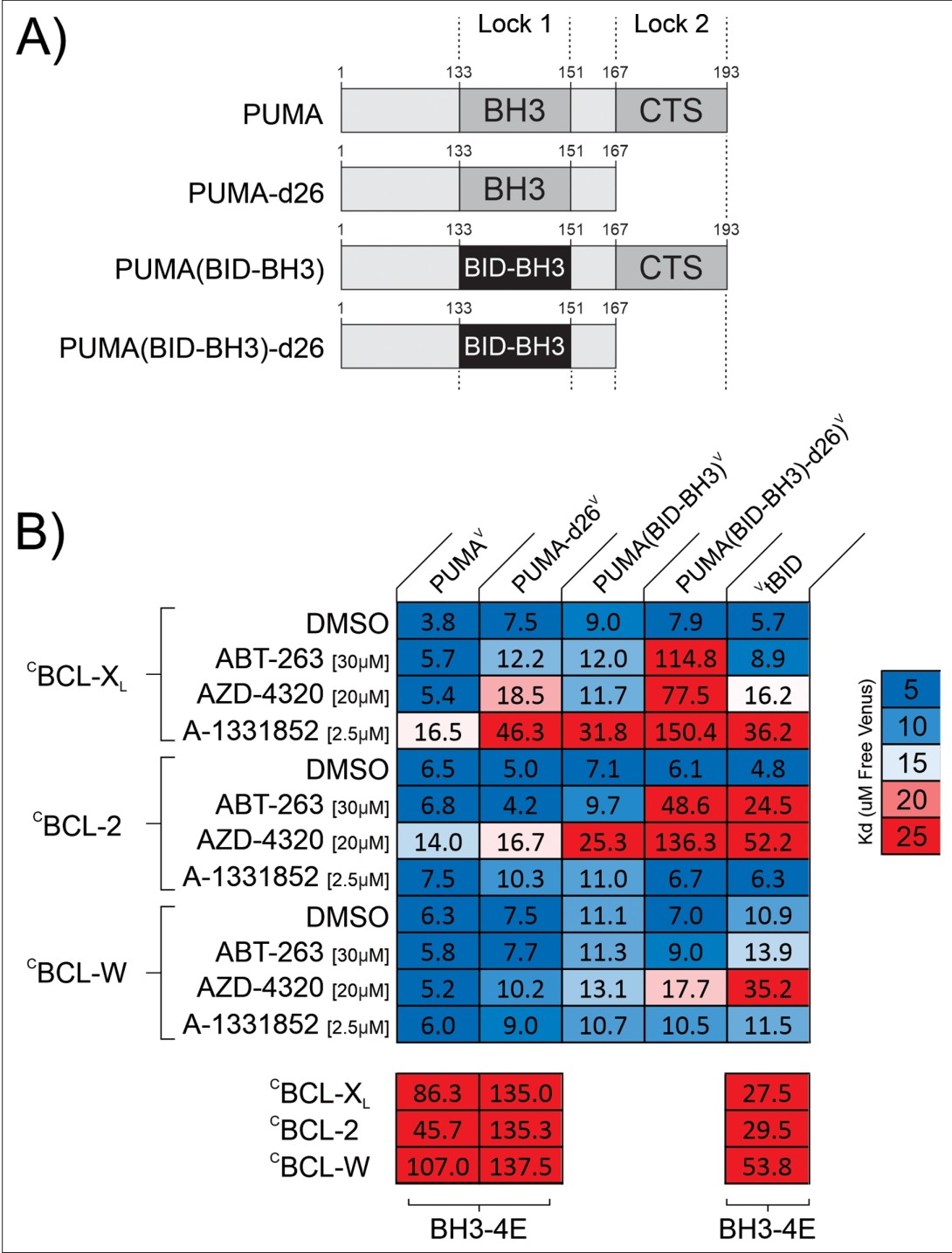

**Figure 3.** PUMA resists BH3-mimetic displacement by double-bolt locking to anti-apoptotic proteins. (**A**) Linear depictions of the PUMA[V] mutants analyzed. Lock 1 indicates the BH3-motif, Lock 2 indicates the CTS. (**B**) Mutation of either the PUMA CTS or the BH3 region is sufficient to relieve resistance to a subset of the drugs resulting in apparent $K_d$ values similar to those obtained for the BH3 mimetic sensitive control [V]tBID. Mutation of both sequences in PUMA results in a protein (PUMA(BID-BH3)-d26[V]) that can be displaced by the BH3 mimetic AZD-4320 from the three anti-apoptotic proteins, [C]BCL-X$_L$, [C]BCL-2, and [C]BCL-W. Heatmaps displaying $K_d$ values calculated from fitting qF[3] binding curves and represented by color (scale to the right) to highlight changes in these binding affinities (binding (blue) to non-binding (red)) within the heatmap cells. The interacting protein pairs are indicated to the left and at the top. Final drug concentrations added are indicated at the left. DMSO is the solvent control. BH3-4E below the panels indicates mutation of the BH3 protein listed at the top. Binding by the BH3-mimetic sensitive control [V]tBID to the anti-apoptotic proteins was inhibited by the cognate BH3-mimetics, as expected. In the assay conditions used, inhibition of [V]tBID indicated that BCL-X$_L$ was inhibited primarily by AZD-4320

*Figure 3 continued on next page*

*Figure 3 continued*

and A-1331852; BCL-2 was inhibited by ABT-263 and AZD-4320; [C]BCL-W was inhibited by only AZD-4320. Data is averages from three independent experiments.

The online version of this article includes the following figure supplement(s) for figure 3:

**Figure supplement 1.** PUMA[V] binding to anti-apoptotic proteins is resistant to displacement by BH3-mimetics in live cells due to the last 26 amino acids in the C-terminus of PUMA.

**Figure supplement 2.** Binding curves generated from qF[3] data demonstrate that both [V]PUMA(BID-BH3) and [V]PUMA(BID-BH3)-d26 bind to [C]BCL-X$_L$, [C]BCL-2, and [C]BCL-W (blue lines).

**Figure supplement 3.** Binding curves generated from qF[3] data demonstrate that [V]tBID binds to [C]BCL-X$_L$, [C]BCL-2, and [C]BCL-W (blue lines) in a BH3-depdendent manner ([V]tBID-BH3-4E, grey lines).

BH3-protein that is easily displaced from anti-apoptotic proteins by BH3-mimetics (*Aranovich et al., 2012*; *Pécot et al., 2016*). The resulting protein, PUMA(BID-BH3)[V] has a compromised BH3-motif, but an intact CTS (*Figure 3A*). To disable both Lock 1 and 2 the CTS was deleted from PUMA(BID-BH3)[V]. The resulting mutant, PUMA(BID-BH3)-d26[V], has a compromised BH3-motif and is missing the PUMA CTS (*Figure 3A*).

Constructs encoding these proteins were transfected into BMK-dko cells expressing one of [C]BCL-X$_L$, [C]BCL-2 or [C]BCL-W, and after ~20 hr, binding interactions at equilibrium in live cells were measured by qF[3]. In the DMSO controls, PUMA[V] bound with K$_d$s of ~4–6 µM to all three anti-apoptotic proteins while the mutants bound with K$_d$s of ~5–10 µM (*Figure 3B*). As expected, addition of BH3-mimetic did not increase the dissociation constants for PUMA[V] substantially, although there was detectable inhibition of BCL-X$_L$ by A-1331852 and BCL-2 by AZD-4320 (*Figure 3B*). Also as anticipated, deletion of the PUMA CTS (PUMA-d26[V]) resulted in increased BH3-mimetic mediated displacement from BCL-X$_L$, but not for BCL-2 and BCL-W as reported above (compare *Figure 2A* and *Figure 3B*). As a positive control for BH3-mimetic mediated displacement, we included Venus fused to the N-terminus of truncated BID ([V]tBID). This protein was displaced from [C]BCL-X$_L$, [C]BCL-2 and [C]BCL-W by the cognate BH3-mimetic to a similar extent as PUMA-d26[V] (*Figure 3B*, far-right column). Replacing the BH3 region of PUMA with that of BID (PUMA(BID-BH3)[V]) was sufficient to result displacement similar to that of [V]tBID from [C]BCL-X$_L$ with the inhibitor A-1331852 and from BCL-2 with AZD-4320. However, in some cases, PUMA(BID-BH3)[V] was more resistant than [V]tBID to displacement by the cognate BH3-mimetics (e.g. [C]BCL-2 with ABT-263 and [C]BCL-W with AZD-4320). In contrast, resistance to displacement from [C]BCL-X$_L$ and [C]BCL-2 was abolished by mutation of Lock 1 and Lock 2 (PUMA(BID-BH3)-d26[V]). While resistance to AZD-4320 was reduced for PUMA(BID-BH3)-d26[V] for binding to [C]BCL-W the apparent K$_d$ remained substantially lower than for [V]tBID. Overall, these data suggest that both the PUMA BH3 and CTS regions contribute to the high affinity interaction of PUMA with anti-apoptotic proteins, and that the primary mechanism of resistance to BH3-mimetics is a 'double-bolt lock', similar to BIM (*Liu et al., 2019*).

## The CTS of PUMA localizes the protein to the endoplasmic reticulum

The data above demonstrate a previously unrecognized function of the PUMA CTS in double-bolt locking to BCL-X$_L$ and BCL-2 and that PUMA-BCL-X$_L$ complexes can form in the absence of membranes. Although, BCL-X$_L$ is found in both the cytoplasm and bound to intracellular membranes, previous reports suggest that PUMA is localized in a CTS dependent manner primarily to mitochondria (*Yu et al., 2001*; *Nakano and Vousden, 2001*; *Wilfling et al., 2012*; *Yee and Vousden, 2008*). However, in these reports, localization was interpreted by visual inspection and was not rigorously analyzed. To our surprise, although the spatial resolution of FLIM-FRET images is limited, [V]PUMA-[C]BCL-X$_L$ complexes in FLIM-FRET images were not obviously localized only at mitochondria (*Figure 2B*). This could be due to complexes forming and remaining cytoplasmic; however, previous reports suggest that binding to membranes increases the stability of the complexes and contributes to the resistance we observed to displacement by BH3 mimetics (*Pécot et al., 2016*). Therefore, we re-examined PUMA localization using confocal microscopy and created mutants to identify the CTS residues that mediate binding to subcellular membranes.

To assign the localization of PUMA in live cells using an unbiased quantitative approach we made use of a previously described random forests classifier built from a reference library of 789,011 optically

validated landmark-based localization images (*Schormann et al., 2020*). Briefly, NMuMG (normal murine mammary gland) cells were infected with lentivirus to express a fusion protein consisting of EGFP fused to the N-terminus of a BH3-4E mutant PUMA to visualize the fusion protein (EGFPPUMA-4E) without it binding to anti-apoptotic proteins or killing the cells. Using automated-confocal microscopy, 2225 images of individual cells expressing EGFPPUMA-4E were classified as the landmark from the reference library they most resemble. To our surprise, 55% of EGFPPUMA-4E cell micrographs were classified as most similar to one of the resident endoplasmic reticulum (ER) markers (blue bars), indicating ER localization (*Figure 4A*). In contrast, smaller fractions of the cells were classified as patterns resembling that of a protein that recycles between the ER and Golgi (Calr-KDEL) or that is resident at mitochondria (MAO, monoamine oxidase). For images of other cells classification was to one of a variety of other locations particularly transport vesicles, a phenomenon not uncommon for overexpressed proteins (*Schormann et al., 2020*). As a positive control for correct classification of an ER localized EGFP-fused BH3-protein, we infected NMuMG cells with lentivirus to express EGFPBIK-L61G a mutant of the tail-anchored ER protein BIK containing the BH3 mutation L61G previously shown to abrogate apoptotic activity (*Mathai et al., 2002*). As expected, 79% of the images of EGFPBIK-L61G expressing cells were classified as showing ER localization (yellow bars) compared to 55% for EGFPPUMA-4E (*Figure 4A*).

As an orthogonal approach to examine localization in the BMK-dko cell line used for FLIM-FRET experiments, co-localization with ER and mitochondrial markers was examined by calculating Pearson's correlation coefficients. Because it lacks both BAX and BAK, the BMK-dko cell line will not undergo apoptosis in response to expression of PUMA or BIK. Therefore, inactivating mutations that might affect localization are not required. To calculate correlation values between PUMA and landmarks for two locations in the same cell, the fluorescent protein mCerulean3 was fused to the N-terminus of PUMA (CPUMA). The ER membrane and mitochondria were visualized using the dyes BODIPY FL thapsigargin (green) and MitoTracker Red, respectively. Individual cells were identified using far red nuclear stain DRAQ5. Images of cells were obtained by automated confocal microscopy (Opera Phenix, PerkinElmer) and analyzed with Harmony software (V4.9). To assess localization to the ER membrane, Pearson's correlation coefficients were calculated between BODIPY FL thapsigargin and mCerulean3 fluorescence intensity for each cell. As seen in *Figure 4B*, the expression of the negative control mCerulean3 alone resulted in a median coefficient close to zero, indicating no correlation. The positive control for ER localization, mCerulean3 fused to the N-terminus of BIK (CBIK), resulted in a median coefficient close to 0.75, suggesting this value represents localization at the ER membrane. Both CPUMA and CPUMA-4E have Pearson's coefficient values with BODIPY FL thapsigargin of ~0.6 similar to what was seen for CBIK. In contrast, the Pearson's correlation coefficient for CPUMA-d26 with BODIPY FL thapsigargin was similar to the negative control (*Figure 4B*) suggesting CPUMA-d26 is located in the cytoplasm. Overall, this data indicates that exogenously expressed PUMA localizes to the ER in a BH3-independent, CTS-dependent manner.

The same approach was used to assess protein localization at the mitochondria. As expected, the Pearson's correlation coefficients calculated between mCerulean3 alone and MitoTracker Red resulted in a median coefficient close to zero, indicating no correlation. As a positive control for perfect mitochondrial localization, the calculated median coefficient between the dyes MitoTracker Green and MitoTracker Red was close to 0.85 (*Figure 4B*). As expected, the median Pearson's coefficient for the ER localized control CBIK with MitoTracker Red was only slightly higher than the cytoplasmic control mCerulean3 alone. The Pearson's correlation coefficients for both CPUMA and CPUMA-4E with MitoTracker Red were less than 0.2, much lower than those calculated for these proteins with the ER marker BIK (*Figure 3B*, compare the bottom and top panels), but slightly higher than values obtained for the cytoplasmic protein mCerulean3. Indeed, the Pearson's correlation coefficients for CPUMA and CPUMA-4E with MitoTracker Red were similar to the values obtained for the ER marker CBIK and MitoTracker Red. Consistent with these observations, the Pearson's correlation coefficients for CPUMA-d26 (lacking the CTS) and mCerulean3 with mitochondria were both close to zero (*Figure 4B*, lower panel). Finally, images of CBCL-X$_L$, a protein known to be both cytoplasmic and localized to multiple membranes resulted in intermediate Pearson's correlation coefficients. Together, this data suggests that when exogenously expressed the PUMA CTS localizes the protein primarily to the ER.

To further dissect the PUMA CTS and identify the region responsible for directing PUMA localization additional mutants were generated. Incremental deletions of the PUMA CTS decreased the

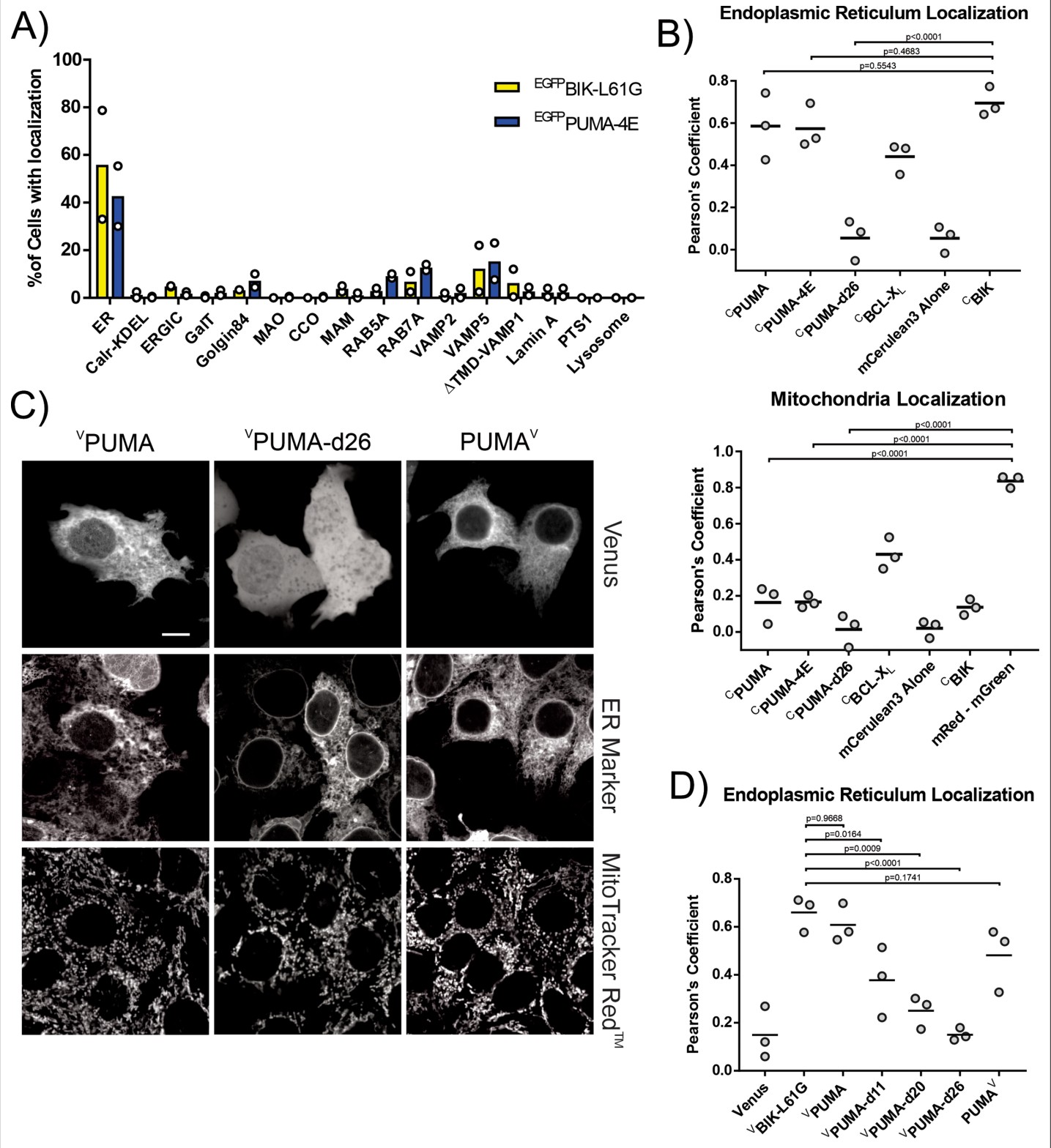

**Figure 4.** The CTS of PUMA localizes primarily to the endoplasmic reticulum. (**A**) Lentiviral infected NMuMG cells expressing EGFP fused BH3-proteins with mutated BH3-motifs to prevent cell death, were imaged using confocal fluorescence microscopy. Images of individual cells were classified as one of the previously defined landmarks as described in *Schormann et al., 2020*. Most cells expressing EGFP PUMA-4E (blue) or the ER localized control EGFP BIK-L61G (yellow) were classified as most closely resembling ER landmarks with a small fraction of cells expressing EGFP PUMA-4E resembling localization in a variety of transport vesicles (RAB5A, RAB7A) or secretory pathway vesicles and plasma membrane (VAMP5). The landmarks tested included ER (3

*Figure 4 continued on next page*

*Figure 4 continued*

resident endoplasmic reticulum membrane markers), Calr-KDEL (recycling between ER and Golgi), ERGIC (ER-Golgi intermediate compartment), GalT (trans-Golgi), Golgin84 (cis-Golgi), MAO (outer mitochondrial membrane), CCO (inner mitochondrial membrane), MAM (mitochondrial associated ER membrane), Rab5A, Rab7A, VAMP2 (transport and secretory vesicles), VAMP5 (Secretory pathway to the plasma membrane), ΔTMD-VAMP1 (cytoplasm), Lamin A (nuclear envelope), PTS1 (peroxisomes), LAMP1 (Lysosomes) (*Schormann et al., 2020*). (**B**) Quantification of fluorescence colocalization in BMK-dko cells indicates PUMA primarily localizes in a CTS-dependent manner to the ER (upper panel) and not Mitochondria (lower panel). Pearson's correlation coefficients from three independent experiments are reported for the indicated proteins with the ER marker BODIPY FL thapsigargin (upper panel) or the mitochondrial marker Mitotracker Red (mRed, lower panel). The horizontal bars indicate the medians and mGreen indicates the stain Mitotracker Green. $^{C}$BIK is an ER marker composed of mCerulean3 fused to the ER localized protein BIK. Data from three independent experiments are shown with horizontal bars indicating the medians. Each data point represents the average from a minimum of 50 cells. (**C**) Fusion of Venus to the C-terminus of PUMA (PUMA$^{V}$) does not prevent localization at the ER, suggesting that the PUMA CTS is not a conventional TA that spans the bilayer. Top row: Micrographs of the Venus fluorescence from cells expressing $^{V}$PUMA, $^{V}$PUMA-d26, and PUMA$^{V}$ by transient transfection, as indicated above. Middle row: Micrographs of the ER marker. Bottom row: MitoTracker Red staining for the same cells. White scale bar is 5 μm. (**D**) Quantification of the extent to which the distribution of the various mutant proteins (indicated below) correlated with the distribution of the ER marker ($^{C}$BIK) in BMK-dko cells. Data from three independent experiments are shown with horizontal bars indicating the medians. Each data point represents the average from a minimum of 50 cells.

The online version of this article includes the following figure supplement(s) for figure 4:

**Figure supplement 1.** Quantification of fluorescence colocalization in BMK-dko cells indicates that fusion of just the PUMA CTS to the carboxyl-terminus of Venus ($^{V}$PUMA-CTS) primarily localizes the fluorescent protein to the ER (left panel) with minimal localization to the Mitochondria, similar to the localization of $^{V}$PUMA (right panel).

**Figure supplement 2.** PUMA binds BCL-XL at ROIs corresponding to mitochondria and ER in live cells.

**Figure supplement 3.** Genotoxic and ER stress inducing drugs cause upregulation of endogenous PUMA protein which localized with mitochondrial and ER markers in MCF-7 cells.

---

Pearson's correlation coefficient of the mutant proteins with ER localization suggesting that deleting as few as the last 11 amino acids ($^{V}$PUMA-d11) impacts ER localization (*Figure 4D*). As progressive deletion of the CTS correlated with decreasing ER localization, this suggests that the entire CTS contributes to PUMA localization. Unexpectedly, the fusion protein with Venus at the C-terminus of PUMA (PUMA$^{V}$) retained ER localization (*Figure 4C and D*), suggesting that unlike CTS of BIK which fully inserts into the ER (*Wilfling et al., 2012*) the CTS of PUMA is not a conventional tail-anchor that spans the membrane bilayer. This conclusion is also congruent with the data indicating that the PUMA CTS sequence is too short to span the ER membrane.

Our data showed that exogenously expressed PUMA mostly localizes to the ER while BCL-X$_{L}$ localized to both mitochondria and ER (*Figure 4B*; *Osterlund et al., 2022*; *Kaufmann et al., 2003*; *Osterlund et al., 2023*). Despite this, in BMK cells, PUMA bound to BCL-X$_{L}$ with an apparent K$_{d}$ of 5.5 μM (*Figure 2*). To further investigate the subcellular localization of PUMA:BCL-X$_{L}$ heterodimers, FLIM-FRET experiments with $^{V}$PUMA were done using MCF-7 cells stably expressing $^{C}$BCL-X$_{L}$ and as a localization marker, mCherry fused to the tail anchor sequence of ActA ($^{Ch}$ActA) or of Cytochrome-b5 ($^{Ch}$Cb5). This enables the differentiation of the mitochondrial ($^{Ch}$ActA) and ER ($^{Ch}$Cb5) subcellular regions within the cells where $^{V}$PUMA could interact with $^{C}$BCL-X$_{L}$ facilitating localization of complexes. The mCherry channel was used to generate a mitochondrial or ER 'mask' and select ROIs for which the mCerulean3 fluorescence lifetime and % FRET were calculated. The plateau observed for the data indicate that both PUMA and PUMA-d26 bind to BCL-X$_{L}$ at both the ER and the mitochondria. In contrast, the FLIM data for the BH3-4E negative controls ($^{V}$PUMA-4E and $^{V}$PUMA-d26-4E) can be fit to a straight-line, indicating collisions (*Figure 4—figure supplement 2B*). The simplest explanation for exogenously expressed $^{V}$PUMA binding to BCL-X$_{L}$ at mitochondria is that BH3-dependent binding of PUMA to mitochondrial localized BCL-XL results in localization of $^{V}$PUMA at or close to mitochondria. Supporting this hypothesis, immunofluorescence staining in MCF-7 cells for endogenous PUMA which is expressed at levels more similar to that of BCL-X$_{L}$ and BCL-2 in these cells (*Antony et al., 2012*; *Mukherjee et al., 2015*) revealed that endogenous PUMA co-localized more with the mitochondrial landmark ($^{Ch}$ActA) (Pearson correlation coefficient ≈ 0.6) than the ER landmark ($^{Ch}$Cb5) (Pearson correlation coefficient ≈ 0.4; *Figure 4—figure supplement 3C*, column 4, 8, and 12).

Given the relatively lower co-localization of endogenous PUMA to the ER marker in MCF-7 cells, we examined localization of endogenous PUMA after the induction of PUMA expression by ER stress (*Reimertz et al., 2003*; *Yu and Zhang, 2008*) or genotoxic stress (*Meyerkord et al., 2008*; *Jamil et al., 2015*). Previous results suggest that a P53-dependent response to genotoxic stress results

in the extension of peripheral tubular ER and promotes the formation of ER-mitochondrial contact sites (*Wang et al., 2007*) which may be binding sites for PUMA. Therefore, to test this hypothesis, immunofluorescence staining for endogenous PUMA was done in [Ch]ActA (mitochondria marker) or [Ch]Cb5 expressing (ER marker) MCF-7 cells treated with Thapsigargin (TG) or Tunicamycin (TN) to induce ER stress and Etoposide (ETOP) to induce genotoxic stress (*Figure 4—figure supplement 3A*). As expected, we observed an increase in the intensity of the Alexa-488 immunofluorescence signal for PUMA in cells treated with the drugs compared to the DMSO-treated cells (*Figure 4— figure supplement 3B*). Intriguingly, we saw an increase in Pearson correlation coefficients for PUMA to both the ER and the mitochondria, most notably at the highest concentrations of the 3 drugs (*Figure 4—figure supplement 3C*). As genotoxic stressors can also induce expression of the anti-apoptotic BCL-X$_L$ (*Jamil et al., 2015*) which is both ER and mitochondria-localized (*Figure 4B*), we speculate that binding to BCL-X$_L$ contributes to increased localization of PUMA with the mitochondrial marker [Ch]ActA. Moreover, stress induced changes in ER structure and increased MAMs would also result in apparent localization at mitochondria. Taken together, these data suggest that PUMA subcellular localization in living cells is dynamic and dependent not only on the inherent specificity of the PUMA CTS but also on the abundance and localization of its binding partners. This phenomenon has been observed with another BH3-only protein, BIK whereby the predominantly-ER localized BIK can localized to the mitochondria in BMK-dko upon expression of a mitochondria-localized BCL-X$_L$ mutant (*Osterlund et al., 2023*).

## Restoring ER localization to PUMA-d26 does not result in BH3-mimetic resistance

Together the data in *Figures 1–3* demonstrate that the CTS of PUMA is required for the protein to resist BH3-mimetic-mediated displacement from BCL-X$_L$ in vitro and in live cells. For purified proteins resistance to BH3-mimetic displacement is independent of binding of the proteins to membranes. However, it remains possible that in live cells, PUMA-d26 no longer resists BH3-mimetic displacement because the protein no longer binds subcellular membranes. To test this hypothesis, we constructed two mutants in which the CTS of PUMA was replaced by the tail-anchor sequences of the two proteins used as ER localized controls: BIK ([V]PUMA-d26-ER1) or cytochrome b5 (CB5) ([V]PUMA-d26-ER2) (*Figure 5A*, *Figure 5—figure supplement 1A*).

To determine if these fusion proteins correctly localize to the ER, they were expressed by transient transfection of the corresponding constructs into BMK-dko cells stably expressing [C]BIK as an ER localization marker (labelled as ER Marker in *Figure 5C*). The cells were stained with the nuclear dye DRAQ5, imaged by automated confocal microscopy and co-localization was assessed using Pearson's correlation coefficients, as described above. Both [V]PUMA-d26-ER1 and [V]PUMA-d26-ER2 generated Pearson's correlation coefficients with the ER localization marker that were substantially greater than those for either [V]PUMA-d26 or Venus alone with the same ER marker (*Figure 5B and C*). In addition to correct localization, exogenous expression of both fusion proteins induced apoptosis in BMK-wt cells, in a BH3-dependent manner, as measured by Annexin V positivity using confocal microscopy (*Figure 5—figure supplement 1B*). Given that the fusion proteins are both functional and correctly localized at ER membranes, we used qF[3] to measure binding with [C]BCL-X$_L$ in live cells. As seen in *Figure 5C*, both [V]PUMA-d26-ER1 and [V]PUMA-d26-ER2 bound to [C]BCL-X$_L$ (DMSO column) in a BH3-dependent manner in BMK-dko cells. However, neither complex was resistant to displacement by the addition of ABT-263, A-1331852 or AZD-4320. Addition of any of these drugs resulted in significantly higher dissociation constants for binding to [C]BCL-X$_L$ for [V]PUMA-d26-ER1 and [V]PUMA-d26-ER2 compared to [V]PUMA. In addition to binding to [C]BCL-X$_L$, [V]PUMA, [V]PUMA-d26-ER1 and [V]PUMA-d26-ER2 also bound to [C]MCL-1 in live BMK-dko cells. Similar to the results obtained for binding to [C]BCL-X$_L$, the addition of MCL-1 specific BH3-mimetics; S63845 and S64315, resulted in higher dissociation constants for binding to MCL-1 by both mutant proteins compared to [V]PUMA (*Figure 5C*). However, these data also demonstrate that the MCL-1 inhibitors only partially displaced [V]PUMA-d26-ER1 and [V]PUMA-d26-ER2 compared to mutation of the PUMA BH3 sequence. Thus, similar to BCL-2 and BCL-W the BH3 sequence of PUMA is sufficient to confer partial resistance of binding to MCL-1 to the drugs.

Thus, these data indicate that reintroducing ER membrane localization to [V]PUMA-d26 does not restore resistance to BH3-mimetic displacement, suggesting that specific residues or sequences within the PUMA CTS contribute to BH3-mimetic resistance.

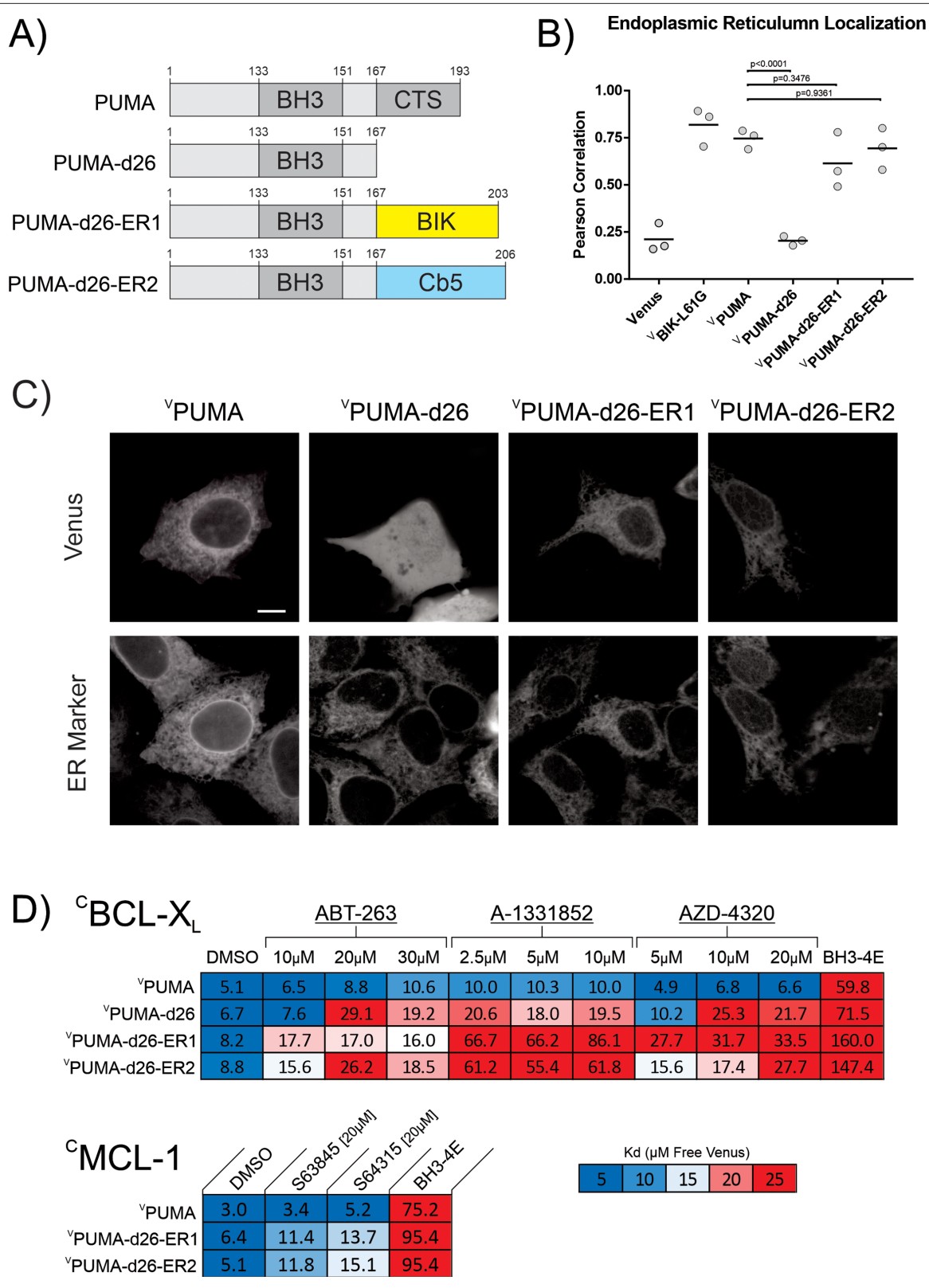

**Figure 5.** Restoring ER localization to PUMA-d26 does not confer resistance to BH3-mimetic displacement. (**A**) Cartoon depiction of fusion proteins created. (**B,C**) Fusion of the canonical tail-anchors from BIK and CB5 to VPUMA-d26 restored ER localization when expressed in BMK-dko cells. (**B**) Pearson's correlation with the ER marker protein CBIK in BMK-dko cells. Data points are averages from independent experiments. A one-way ANOVA and Dunnett's multiple comparisons test were used to calculate the indicated p-values. (**C**) Micrographs illustrating subcellular localization by

*Figure 5 continued on next page*

Figure 5 continued

confocal microscopy of the indicated Venus fusion proteins co-expressed with the ER marker protein [C]BIK. The scale bar indicates 5 µm. (**D**) Heatmaps generated from qF3 data display calculated apparent $K_d$'s for binding of the indicated mutants to [C]BCL-XL and [C]MCL-1 in live BMK-dko cells. Restoring ER localization to PUMA-d26 ([V]PUMA-d26-ER1 and [V]PUMA-d26-ER2) did not restore resistance to BH3-mimetic displacement as indicated by increased dissociation constants in the presence of BH3-mimetic.

The online version of this article includes the following figure supplement(s) for figure 5:

**Figure supplement 1.** Exogenous expression of [V]PUMA and [V]PUMA-ER localized tail-anchor chimera mutants induce apoptosis in BMK cells.

**Figure supplement 2.** Binding curves generated from qF[3] data demonstrate that [V]PUMA, [V]PUMA-d26, [V]PUMA-d26-ER1, and [V]PUMA-d26-ER2 bind to [C]BCL-X[L] (blue lines) in a BH3-dependent manner (BH3-4E, grey lines, indicate primarily collisions).

**Figure supplement 3.** Binding curves generated from qF[3] data demonstrate that [V]PUMA, [V]PUMA-d26, [V]PUMA-d26-ER1 and [V]PUMA-d26-ER2 bind to [C]BCL-X[L] (blue lines) in a BH3-dependent manner (BH3-4E, grey lines, indicate primarily collisions).

**Figure supplement 4.** Binding curves generated from qF[3] data demonstrate that [V]PUMA, [V]PUMA-d26-ER1 and [V]PUMA-d26-ER2 bind to [C]MCL-1 (blue lines) in a BH3-dependent manner (BH3-4E, grey lines).

## Residues I175 and P180 in the PUMA CTS contribute to both ER localization and BH3-mimetic resistance

Data with the CTS substitutions above indicated that localization at the ER is not sufficient to confer resistance to BH3 mimetics. Thus, we sought to identify which regions and residues within the CTS of PUMA are required to confer resistance to BH3 mimetics and if such regions/residues are also required for membrane binding. To this end, mutants were generated containing serial deletions of the PUMA CTS sequence. These mutants were then used to measure binding to membranes and to [C]BCL-X[L] in live cells using qF[3]. Deletion of the last 11 residues ([V]PUMA-d11) had no effect on localization, binding to BCL-X[L] or resistance of the complex to BH3-mimetic inhibition (apparent $K_d$'s: DMSO = 6 µM, plus 2.5 mM A-1331852=14 µM) (*Figure 6A–C*). Deletion of the last 20 residues ([V]PUMA-d20) negatively impacted localization (*Figure 4D*) and substantially altered binding ($\Delta\omega$). However, the data still resemble a binding curve and although there is substantial noise at low free Venus concentrations where there was no clear change in resistance to addition of BH3-mimetic (apparent $K_d$'s: DMSO = 2 µM, plus 2.5 mM A-1331852=13 µM) (*Figure 6B*). The observed change in $\Delta\omega$ could be due to a conformational change in the protein that increased the distance or altered the dipole orientations between the FRET donor ([C]BCL-X[L]) and acceptor ([V]PUMA-d20) rather than a change in binding affinity. In contrast, deletion of the last 26 amino acids of PUMA ([V]PUMA-d26) eliminated both ER localization and BH3 mimetic resistant binding to [C]BCL-X[L] (apparent $K_d$'s: DMSO = 8 µM, plus 2.5 mM A-1331852=38 µM). Therefore, the residues between [V]PUMA-d11 and [V]PUMA-d20 affect localization and $\Delta\omega$, but the effect on BH3 mimetic resistance is less certain. Moreover, this region of the PUMA CTS is the most hydrophobic and contains two proline residues which are unusual for a membrane binding region. Therefore, to probe this region of the protein, nine mutants of PUMA[V] were generated in which individual residues were replaced with glutamic acid (E). Although replacement with E is a non-conservative mutation, for the DMSO controls similar dissociation constants were measured for all the mutants of PUMA[V] for binding to [C]BCL-X[L] (*Figure 6C*). Therefore, none of the mutations abolished binding of PUMA to BCL-X[L]. Addition of the potent BCL-X[L] inhibitor A-1331852 resulted in an increase in the apparent $K_d$'s for all the mutants. However, for I175E and P180E, the $K_d$'s increased substantially, indicating displacement from BCL-X[L] equivalent to that observed for PUMA-d26[V] (*Figure 6C*).

Visual inspection supported by Pearson's correlation coefficients revealed that PUMA[V] mutants I175E and P180E were not localized to membranes but were located primarily in the cytoplasm (*Figure 6D* and *Figure 6—figure supplement 3*). In contrast, and consistent with the effects seen on binding to [C]BCL-X[L], the other point mutations reduced PUMA localization at ER to a similar minor extent (*Figure 6D*). Therefore, in live cells residues I175 and P180 in the PUMA CTS appear to be required for both binding to membranes and resisting BH3-mimetic displacement. For the mutations with minor effects, there is also a correlation between membrane binding and BH3-mimetic resistance.

Substitution of amino acids in the PUMA CTS with negatively charged E residues are more likely to influence PUMA binding to membranes than more conservative substitutions such as alanine. Therefore, to determine if more conservative mutations would separate membrane binding from resistance to BH3-mimetic, additional PUMA[V] mutants I175A and P180A were analyzed for binding to [C]BCL-X[L] by FLIM-FRET and for localization by Pearson's correlation with an ER marker in live BMK-dko cells.

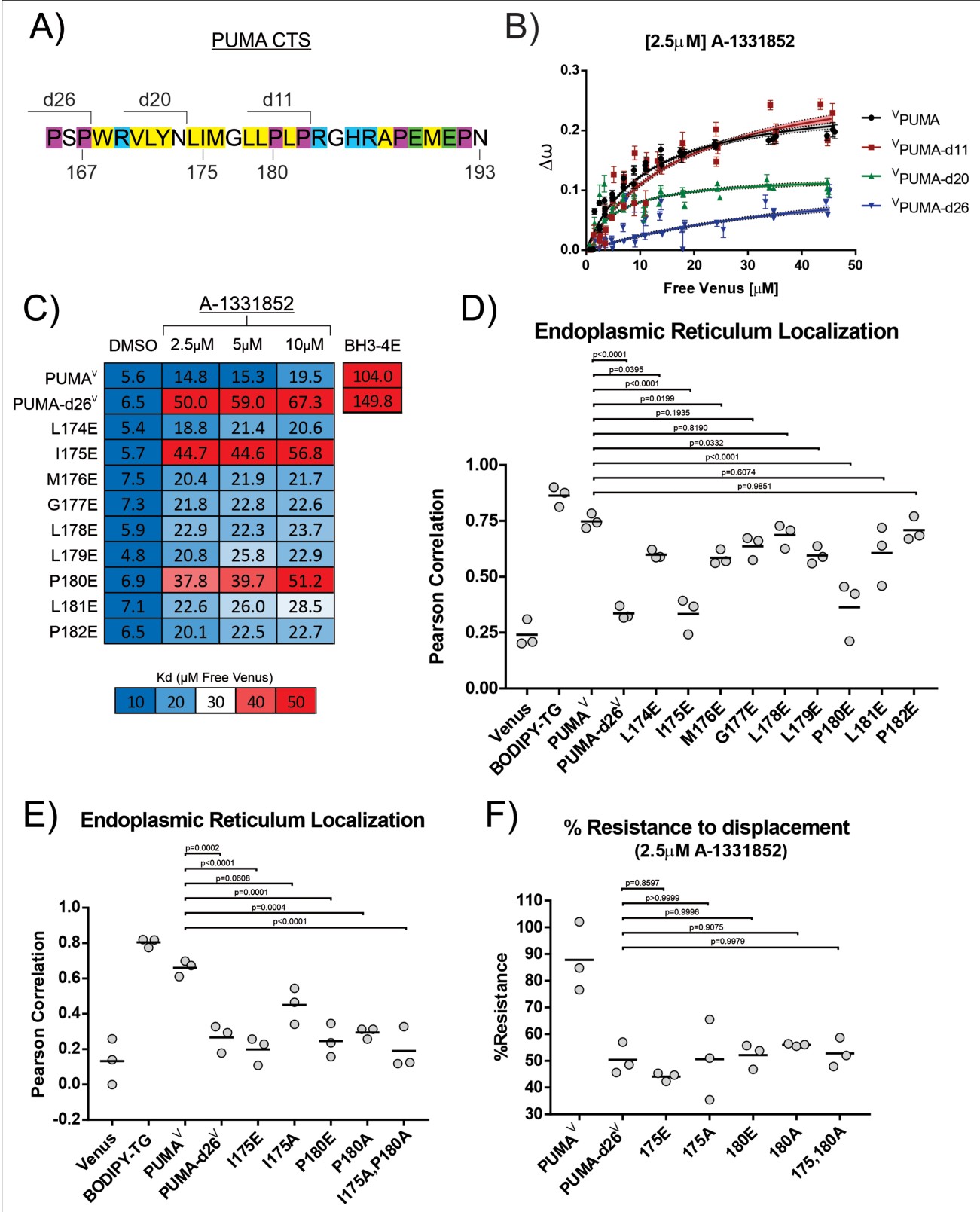

**Figure 6.** Residues I175 and P180 in the PUMA CTS contribute to both ER localization and BH3-mimetic resistance. (**A**) Amino acid composition of the PUMA CTS. Color depicts amino acid chemical properties (yellow = hydrophobic, purple = alpha-helix breaking residue, blue = positively charged, green = negatively charged). Lines indicate deletion points. Residue numbers are indicated below the sequence. (**B**) Binding curves generated by qF[3] for the indicated mutants. [V]PUMA and [V]PUMA-d11 resisted BH3-mimetic displacement from BCL-X$_L$ while [V]PUMA-d26 was displaced (higher K$_d$

*Figure 6 continued on next page*

*Figure 6 continued*

value). $^V$PUMA-d20 binding was at least altered such that the distance between the donor and acceptor was increased, as shown by the dramatically reduced values for $\Delta\omega$, a measure directly related to FRET efficiency and bound fraction. (**C**) Calculated $K_d$ values determined by qF$^3$ for PUMA$^V$ mutants containing a single-glutamic acid substitution in the PUMA CTS displayed as a heat map with calculated values in the heatmap cells. Averages from 3 biological replicates are shown and suggest that residues I175 and P180 are required for resistance to BH3-mimetics. The color scale was changed for this figure to visually differentiate the effect of the point mutations I175E and P180E from the change in binding that resulted from substituting residues at other locations with a glutamic acid residue. (**D**) Pearson's correlation coefficients calculated from confocal micrographs for co-localization of PUMA$^V$ and the indicated PUMA$^V$ mutants with the ER localization marker $^C$BIK suggest that residues I175 and P180 are most important for PUMA localization at the ER. (**E**) Pearson's correlation coefficients calculated from confocal micrographs for co-localization indicate a more conservative mutation to alanine at position I175 (PUMA$^V$ I175A) results in increased localization to the ER, while PUMA$^V$ P180A was not localized at membranes. (**F**) Mutation of residues I175 and P180 abrogated resistance to displacement by the BH3-mimetic A1331852 equivalent to deletion of the entire CTS. % Resistance to displacement of PUMA$^V$ mutants (indicated below) from $^C$BCL-X$_L$ calculated from FLIM-FRET binding curves. Data points are averages from independent experiments. Line indicates the mean of the data points shown. P values in panels (**D,E,F**) were calculated using an ordinary one-way ANOVA method GraphPad Prism 9.5.0 to examine the differences in the mean Pearson Correlation Coefficient values between the tested group and the reference groups (PUMA$^V$ for panels D and E, PUMA-d26$^V$ for panel F).

The online version of this article includes the following figure supplement(s) for figure 6:

**Figure supplement 1.** Binding curves generated from qF$^3$ data demonstrate that PUMA$^V$ and PUMA-d26$^V$ bind to $^C$BCL-X$_L$ (blue points) in a BH3-dependent manner (BH3-4E, grey points).

**Figure supplement 2.** Binding curves generated from qF$^3$ data demonstrate that PUMA$^V$ mutants bind to $^C$BCL-X$_L$ (blue points) in a BH3-dependent manner (negative control, BH3-4E, grey points).

**Figure supplement 3.** Representative confocal micrographs of the Venus fluorescence from BMK-dko cells expressing the proteins indicated above or the PUMA$^V$ mutants indicated by the amino acid substitution.

PUMA$^V$ mutant P180A appeared cytoplasmic, and resulted in a Pearson's coefficient with the ER-localized control that was similar to that of the Venus cytoplasmic control (*Figure 6E* and *Figure 6—figure supplement 3*). This indicates residue P180 is required for PUMA binding to membranes in live cells. In contrast, mutant I175A appears membrane localized upon visual inspection, and generated a calculated Pearson's coefficient with the ER-localized control that was higher than that of either PUMA$^V$ I175E or Venus (*Figure 6E* and *Figure 6—figure supplement 3*), but that was lower than the correlation between the ER-localized control and wildtype PUMA$^V$ (*Figure 6E*).

To compare the extent to which these mutants resist BH3-mimetic displacement from BCL-X$_L$ in live cells, FLIM-FRET measurements were made and % Resistance was calculated as previously described *Liu et al., 2019*; *Pemberton et al., 2019*. Briefly, the FLIM-FRET efficiency at the acceptor/donor ratio of two was calculated from the fitted Hill-slope equation for expression of the proteins alone and in the presence of the BH3-mimetic, and the change in these values was used to calculate the % Resistance (ie the % of complexes that remain intact). Therefore, a low % Resistance is indicative of protein displacement by BH3-mimetic treatment. In the presence of 2.5 µM A-1331852, PUMA$^V$ has a high calculated % Resistance (~90%), while PUMA-d26$^V$ has a lower % Resistance (~50%) (*Figure 6F*). PUMA$^V$ mutants I175E, I175A, P180E and P180A all have low % Resistance values similar to PUMA-d26$^V$, suggesting that all of the mutant PUMA$^V$ proteins were displaced from BCL-X$_L$ similarly to PUMA-d26$^V$. Even PUMA$^V$ I175A that localized partially to the ER (*P*=0.04) did not resist displacement (*Figure 6E–F*). Overall, this data suggests that PUMA CTS residues I175 and P180 are required for both PUMA$^V$ binding to membranes and for resisting BH3-mimetic induced displacement from $^C$BCL-X$_L$.

## Discussion

The availability of small-molecule inhibitors for anti-apoptotic proteins has generated new insights into the mechanisms underlying programmed cell death in a number of cell types, and has positively impacted the treatment of cancer (*Delbridge et al., 2016*). The development of the first true BH3-mimetic, ABT-737, was based on the structure of the BH3 sequence of BAD bound to BCL-X$_L$ and efficacy defined by displacement of BAD BH3-peptides from truncated BCL-X$_L$ in solution (*Oltersdorf et al., 2005*). The other BH3-mimetics currently under pharmaceutical development were all derived based on the structures of BH3-peptides and their corresponding binding sites in the different anti-apoptotic proteins. However, other factors that contribute to BCL-2 family interactions have been discovered. For instance, the MOM and the ER membrane act as both the platform and an active

participant in BCL-2 family interactions (*Pécot et al., 2016*; *Lovell et al., 2008*; *Bleicken et al., 2017*). Not surprisingly, full-length proteins have different affinities than truncated proteins binding to BH3 peptides (*Kale et al., 2018*). This is most dramatically shown for the BH3 protein BIM binding to BCL-X$_L$ (*Liu et al., 2019*; *Chi et al., 2020*). Studies with truncated protein and BH3 peptides showed binding affinities in the micromolar to millimolar range while studies with full length proteins in the presence of membranes revealed a nanomolar affinity between BIM and BCL-X$_L$ (see also *Figure 1*). In the latter studies, both BH3 regions and the CTS previously thought exclusively involved in binding of BIM to membranes were discovered to contribute to BIM function. Moreover, both the BH3 sequence and the CTS are required for BIM to bind BCL-X$_L$, forming a "double-bolt lock", in a manner that is resistant to inhibition by BH3-mimetics (*Liu et al., 2019*).

Collectively the data presented here demonstrate that like BIM, the PUMA CTS functions in binding the protein to membranes and together with the BH3 sequence is required for BH3-mimetic resistant binding to anti-apoptotic proteins. However, unlike BIM, the CTS of PUMA localizes primarily at ER instead of mitochondria (*Figure 4*) and together with the PUMA BH3 region results in almost complete resistance to all currently known BH3 mimetics (*Figure 5D*). Endogenous PUMA was observed predominantly at mitochondria in MCF-7 cells (*Figure 4—figure supplement 3*), suggesting other factors besides the CTS may influence PUMA localization, such as the presence of PUMA-binding partners including the anti-apoptotic protein BCL-X$_L$ at both the ER and mitochondria (*Figure 4B* and *Figure 4—figure supplement 2*).

Unexpectedly, when analyzed using purified proteins the CTS of PUMA conferred resistance to BH3 mimetics in the absence of membranes, demonstrating that inhibition of anti-apoptotic proteins by PUMA does not require binding to membranes (*Figure 1*) and suggesting the PUMA CTS binds to BCL-X$_L$. Furthermore, when the PUMA CTS sequence was replaced with classical tail-anchor sequences from each of two other proteins, the mutant proteins were localized at the ER in live cells but remained sensitive to displacement from BCL-X$_L$ by BH3 mimetics (*Figure 5*). Thus, we conclude that membrane binding is not sufficient for the PUMA CTS to confer resistance to displacement from BCL-X$_L$ by BH3 mimetics. Instead binding of both the BH3 region and CTS of PUMA to BCL-X$_L$ increases both the affinity and avidity of the interaction.

To our surprise delineating the specific residues involved in resistance to BH3 mimetics and localization of PUMA at membranes by mutagenesis revealed that residues I175 and P180 are required for both localization and resistance to BH3-mimetics. This suggests that, at least in live cells, binding to membranes and resistance to BH3-mimetics may not be separable. This is very different from the CTS of BIM in which separate regions of the CTS are involved in binding BIM to mitochondria and binding it to BCL-X$_L$ (*Liu et al., 2019*). It remains to be determined how residues I175 and P180 are involved in PUMA binding to both membranes and anti-apoptotic proteins. We speculate that for P180E or P180A mutation, the diminished membrane binding and resistance to BH3-mimetic displacement might be due to disruption of the structure of the PUMA CTS. As the CTS sequence contains 4 prolines in the last 14 amino acids and there are 6 prolines in the last 30 amino acids of PUMA the structure of this region is expected to be very unlike other tail-anchor sequences that localize proteins at the ER or mitochondria (*Figure 5—figure supplement 1*).

Unexpectedly, we find using two different quantitative approaches that the PUMA CTS localizes the protein primarily to the ER, and not the MOM as previously claimed (*Wilfling et al., 2012*). Random forest image classification is an unbiased approach to assess the localization of $^{EGFP}$PUMA-4E in live NMuMG cells (*Figure 4A*). A caveat of this approach is that it assumes that the 4E mutation that disrupts the BH3 region and abrogated pro-apoptotic function has no effect on PUMA subcellular localization. Therefore, since binding to anti-apoptotic proteins such as BCL-X$_L$ and MCL-1 requires an intact BH3 region, localization due to binding these proteins would not be observed using $^{EGFP}$-PUMA-4E. For this reason, we also calculated Pearson's Correlation values for the overlap between exogenously expressed $^C$PUMA or $^V$PUMA, with ER membrane and mitochondrial markers in live BMK-dko cells (*Figure 4B, C and D*). Moreover, fusion of the PUMA CTS (residues 167–193 of PUMA) to the carboxyl-terminus of the fluorescent protein Venus ($^V$PUMA-CTS) and expression of this mutant in BMK-dko cells results primarily in ER localization (*Figure 4—figure supplement 1*), indicating that the CTS of PUMA alone is sufficient to direct ER localization. Although we have shown that PUMA with an N-terminal fluorescence protein for visualization is functional as a sensitizer and can activate BAX or BAK indirectly to kill cells, it remains possible that the absence of available pro-apoptotic

protein binding partners (BAX and BAK) may alter PUMA localization. Nevertheless, taken together the two approaches provide strong evidence that in epithelial cells, in the absence of cellular stress, the majority of exogenously expressed PUMA is located at the ER. In contrast, endogenous PUMA was located primarily at mitochondria and when expression increased in response to stress agonists, PUMA localization was increased at both ER and mitochondria. We speculate that the localization of PUMA may be determined by the availability and localization of anti-apoptotic binding partners such as BCL-X$_L$. It is also possible that the protein localizes at ER-mitochondrial contact sites similar to the BH3-protein BIK (*Osterlund et al., 2023*) or that localization at the ER may be required for a non-apoptotic function of the protein, sometimes referred to as a 'day job'.

Our data demonstrate that PUMA residues I175 and P180 are required for BH3-mimetic resistance of PUMA$^V$ binding to $^C$BCL-X$_L$. However, in the absence of a BH3-mimetic, the entire CTS region is dispensable for binding of the two proteins as shown by binding of PUMA-d26$^V$ to $^C$BCL-X$_L$ (*Figure 6C*). In contrast, disabling the BH3 region of PUMA$^V$ (BH3-4E mutation) resulted in no binding to anti-apoptotic proteins (*Figure 6—figure supplement 1*). This may suggest that a conformational change in either PUMA or the anti-apoptotic proteins that results from binding of the PUMA BH3 sequence increases the affinity of the interaction with the PUMA CTS. Further delineation of the molecular mechanism may require high resolution structure determination. Nevertheless, our data using purified proteins clearly demonstrate that high affinity binding of PUMA conferred by the CTS occurs in solution and strongly suggests direct binding of the PUMA CTS to BCL-X$_L$ is distinct from the BH3-binding site. Thus, PUMA CTS binding would increase both the affinity and avidity of the interaction via a mechanism that does not require PUMA binding to membranes (*Figure 1*). That BH3-mimetic resistant binding requires only PUMA and $^C$BCL-X$_L$ in solution suggests that future structural studies may be possible.

Finally, our demonstration that PUMA-d26 binding to BCL-2 is more resistant to inhibition by BCL-2 specific BH3-mimetics suggests that the BH3 sequence of PUMA makes unique contacts with BCL-2 not seen with other BH3-proteins. Only when the BH3-motif is changed to a susceptible one (BID-BH3) and the CTS is deleted (PUMA(BID-BH3)-d26$^V$) can BH3-mimetic treatment fully inhibit PUMA binding to BCL-2 (*Figure 3B*). Thus, it may be possible to design BH3-mimetics selective for inhibition of PUMA binding to specific anti-apoptotic proteins. PUMA(BID-BH3)$^V$ that contains the BH3-mimetic displaceable BH3 sequence and the PUMA CTS was more resistant than $^V$tBID, to displacement by the cognate BH3-mimetics (e.g. $^C$BCL-2 with ABT-263 and $^C$BCL-W with AZD-4320). Together these results suggest a role for the PUMA CTS in binding to anti-apoptotic proteins that depends on BH3 sequence binding but is independent of the specific BH3 sequence. We speculate that binding of a BH3 sequence may lead to a conformational change in the anti-apoptotic protein that facilitates binding of the PUMA CTS. Finally, the unique amino acid sequence of the PUMA CTS argues that generating an efficient specific inhibitor of PUMA binding to anti-apoptotic proteins may require inhibiting binding of both the PUMA BH3-motif and its CTS.

## Materials and methods

**Key resources table**

| Reagent type (species) or resource | Designation | Source or reference | Identifiers | Additional information |
|---|---|---|---|---|
| Antibody | Antibody to PUMA (polyclonal) | Cell Signaling Technology | Cat. #: 4976 S; RRID: AB_2064551 | Dilution (1:400) |
| Antibody | Alexa-488 anti-Rabbit IgG secondary antibody (polyclonal) | Abcam | Cat. # ab150077; RRID: AB_2630356 | Dilution (1:1000) |
| Cell line (*M. musculus*) | Baby Mouse Kidney (BMK)- WT | PMID:11836241 | | Dr. Eileen White (Rutgers University) |
| Cell line (*M. musculus*) | Baby Mouse Kidney (BMK)- DKO | PMID:11836241 | | Dr. Eileen White (Rutgers University) |
| Cell line (*M. musculus*) | NMuMG | | RRID: CVCL_0075 | Dr. Jeff Wrana (University of Toronto) |

*Continued on next page*

*Continued*

| Reagent type (species) or resource | Designation | Source or reference | Identifiers | Additional information |
|---|---|---|---|---|
| Cell line (*H. sapiens*) | HEK293 | Other (*Graham et al., 1977*) | RRID: CVCL_0045 | Provided by Dr. Frank Graham (McMaster University). |
| Cell line (*H. sapiens*) | HCT-116 | Other (*Polyak et al., 1996*) | RRID: CVCL_0291 | Provided by Dr. Bert Vogelstein (John Hopkins University). |
| Cell line (*H. sapiens*) | MCF-7 | PMID:3790748 | RRID: CVCL_0031 | Provided by Dr. Ronald N. Buick (University of Toronto) |
| Chemical compound, drug | DRAQ5 | ThermoFisher Scientific, Molecular probes | Cat. #62251 | Far red nucleic acid specific fluorescent dye for cell imaging |
| Chemical compound, drug | MitoTracker Red | ThermoFisher Scientific, Molecular probes | Cat. #: M22425 | Mitochodria specific fluorescent dye |
| Chemical compound, drug | BODIPY FL thapsigargin | MedChemExpress | Cat. #: HY-D1608 | ER specific fluorescent dye |
| Chemical compound, drug | Alexa 647-maleimide | ThermoFisher Scientific, Molecular probes | Cat. #: A20347 | Thiol reactive fluorescent dye |
| Chemical compound, drug | Alexa568-maleimide | ThermoFisher Scientific, Molecular probes | Cat. #. A20341 | Thiol reactive fluorescent dye |
| Chemical compound, drug | PC (L-α-phosphatidylcholine) | Avanti Polar Lipids | Cat. #:840051 C | for making liposomes, used 48% PC |
| Chemical compound, drug | DOPS (1,2-dioleoyl-sn-glycero-3-phospho-L-serine) | Avanti Polar Lipids | Cat. #: 840035 C | for making liposomes, used 10% DOPS |
| Chemical compound, drug | PI (L-α-phosphatidylinositol) | Avanti Polar Lipids | Cat. #:840042 C | for making liposomes, used 10% PI |
| Chemical compound, drug | PE (L-α-phosphatidylethanolamine) | Avanti Polar Lipids | Cat. #: 841118 C | for making liposomes, used 28% PE |
| Chemical compound, drug | TOCL, (18:1 Cardiolipin) | Avanti Polar Lipids | Cat. #: 710335 C | for making liposomes, used 4% TOCL |
| Chemical compound, drug | A-1331852 | Chemietek | Cat. #: CT-A115 | in DMSO |
| Chemical compound, drug | S-63845 | Chemietek | Cat. #: 1799633-27-4 | in DMSO |
| Chemical compound, drug | S-64315 also named "MIK665" | ChemieTek | Cat. # CT-MIK665 | in DMSO |
| Chemical compound, drug | Navitoclax; ABT-263 | Selleckchem | Cat. #: S1001 | in DMSO |
| Chemical compound, drug | AZD4320 | ChemieTek | Cat. #: CT-A4320 | in DMSO |
| Chemical compound, drug | Thapsigargin | Sigma-Aldrich | Cat. #:T9033 | in DMSO |
| Chemical compound, drug | Etoposide | Sigma-Aldrich | Cat. #: 33419-42-0 | in DMSO |
| Chemical compound, drug | Tunicamycin | Sigma-Aldrich | Cat. #: T7765 | in DMSO |
| Commercial assay or kit | TransIT-X2 | Mirus | Cat. #: Mir 6003 | Transfection reagent |
| Other | Cell Carrier-384, Ultra | PerkinElmer | Cat. #: 6057300 | for live cell imaging |
| Other | Non-binding surface, 96-well plate, black with clear bottom | Corning | Cat. #: 3881 | For recombinant protein and liposome assays critical to use non-binding plate |

*Continued on next page*

*Continued*

| Reagent type (species) or resource | Designation | Source or reference | Identifiers | Additional information |
|---|---|---|---|---|
| Other | Opera Phenix | PerkinElmer | Cat. #: HH14000000 | Automated spinning disc confocal microscope |
| Other | INO-FHS microscope | PMID:35442739 | | Custom built by INO for Dr. David Andrews' lab |
| Other | The Infinite M1000 | Tecan | | Platereader for in vitro assays with recombinant proteins and liposome assays |
| Gene (*H. sapiens*) | Bax | PMID:14522999, | GI: L22473.1 | For recombinant protein |
| Gene (*H. sapiens*) | Bcl-XL | PMID:18547146 | GI: Z23115.1 | For recombinant protein |
| Gene (*H. sapiens*) | PUMA | PMID:35442739 | GI: 27113 Addgene plasmid# 166739 | |
| Gene (*M. musculus*) | BimL | PMID:30860026 | GI: AAD26594.1 | For recombinant BimL purification |
| Gene (*M. musculus*) | tBid | PMID:22464442 | GI: NM_007544.4 | for expression of VtBid in cells |
| Gene (*M. musculus*) | Bid | PMID:19062087 | GI: NM_007544.4 | Jean-Claude Martinou, SeronoPharmaceutical Institutte For purification of Bid |
| Software, algorithm | GraphPad Prism, version 6 | San Diego, California | RRID: SCR_002798 | Scientific graphing program, used to perform statistical analysis |
| Software, algorithm | MATLAB with toolboxes: Signal Processing, Curve Fitting, Image Processing, Version R2020a | https://doi.org/10.5683/SP3/ZKXQW8 | RRID:SCR_00162 | https://www.mathworks.com/products/matlab.html |
| Software, algorithm | INO software package including INO-FHS Acquisition, INO_FHS_Analysis, and INO_FHS_Batch Analysis | | INO Client Release_r10357 package | |
| Transfected Construct | mVenus-pEGFP-C1 | other | GI: KU341334.1 | Dr. Ray Truant (McMaster University). Backbone EGFP-C1 (Clonetech) |

## Purification of proteins

Full length human BCL-X$_L$ and human BAX were purified as previously described (*Pogmore et al., 2016*). Full length PUMA(α isoform) was expressed in Bl21-AI *Escherichia coli* purchased from New England Biolabs, pelleted using centrifugation, then lysed with the following buffer: *PUMA Lysis Buffer* – pH 8.5, 20 mM HEPES, 50 mM NaCl, 0.2% w/v CHAPS, 20% w/v Glycerol, 50 mM Imidazole, 10 µg/mL DNase, 1 mM PMSF. The lysate was then centrifuged to pellet debris. The supernatant was loaded onto a chitin resin column, 2 mL total volume (New England Biolabs) pre-equilibrated with *PUMA Lysis Buffer*. The column was then washed with *PUMA Chitin Wash Buffer* - pH 8.5, 20 mM HEPES, 50 mM NaCl, 0.6% w/v CHAPS, 20% w/v Glycerol, 50 mM Imidazole. After washing, 10 mL of *PUMA Chitin Cleavage Buffer* was poured on top the resin allowing ~8 mL to flow through. *PUMA Chitin Cleavage Buffer* - pH 8.5, 20 mM HEPES, 10 mM NaCl, 1% w/v Triton-X-100, 20% w/v Glycerol, 50 mM Imidazole, 300 mM Hydroxylamine. The column was capped retaining ~2 mL of the cleavage buffer on top of the chitin resin and left at 4 °C for 48–72 hours.

For labelling PUMA with a fluorescent dye,~2 mL was eluted from the chitin column, and the hydroxylamine was removed by gel filtration on a 10 mL Sephadex G25 medium column (GE Healthcare life science) equilibrated with *PUMA Chitin Cleavage Buffer* without hydroxylamine. The concentration of the eluted PUMA (~0.5 mL) was determined by spectrophometry and then labeled by adding urea to a final concentration of 3 M, 4 x molar excess relative to total protein of TCEP and 20 x molar excess of AlexaFluor 568 C$_5$ Maleimide (ThermoFisher Scientific). The labelling reaction was adjusted to pH 7.2 using 12 M HCl, and rotated at 37 °C for 3 hours. The ~0.5 mL labelling reaction or chitin column elution (for unlabeled protein) was diluted in 30 mL of *PUMA Ni-Binding Buffer* - pH 7.2,

20 mM HEPES, 100 mM NaCl, 1% w/v CHAPS, 20% w/v Glycerol, 15 mM Imidazole and loaded onto a 0.5 mL HisPur Ni-NTA resin column (ThermoFisher Scientific). After washing the Ni-NTA resin with *PUMA Ni-Wash Buffer 1 p*H 7.2, 20 mM HEPES, 50 mM NaCl, 0.4% w/v CHAPS, 20% w/v Glycerol, 20 mM Imidazole and then with *PUMA Ni-Wash Buffer 2 p*H 7.2, 20 mM HEPES, 50 mM NaCl, 20% w/v Glycerol, 20 mM Imidazole, 0.5 M guanidine hydrochloride (GdnHCl) PUMA protein was eluted from the Ni-NTA resin with *PUMA Ni-Elute Buffer -* pH 7.2, 20 mM HEPES, 10 mM NaCl, 20% w/v Glycerol, 300 mM Imidazole, 1 M GdnHCl. Protein was quantified using a Bradford assay, flash frozen and stored at minus 80 °C until use.

## SMAC-mCherry Mitochondria Release Assay

SMAC-mCherry release assays were carried out as previously described (*Niu et al., 2017*). Briefly, BAX$^{-/-}$/BAK$^{-/-}$ baby mouse kidney (BMK) cells stably expressing a fusion protein comprised of the mitochondrial import peptide of SMAC (amino acids 1–58) fused to the amino-terminus of the fluorescent protein mCherry were lysed by nitrogen cavitation at 150 psi for 10 min on ice in buffer containing 20 mM HEPES (pH 7.2), 250 mM sucrose, 150 mM potassium chloride, 1 mM EDTA, 1 X protease inhibitor cocktail. Nuclei and cell debris were removed by centrifugation of the lystate at 2000 g for 4 min at 4 °C. Heavy membranes containing mitochondria were obtained by centrifuging the supernatant at 13,000 g for 10 min at 4 °C and washing the pellet once in lysis buffer. Membrane fractions (0.2 mg/ml protein) were incubated with desired BCL-2 family proteins in 250 µL volume and incubated for 30 min at 37 °C. Mitochondria were then pelleted by centrifugation at 13,000 g for 10 min. The release of SMAC-mCherry was measured as fluorescence intensity using a Tecan M1000 microplate reader and comparing fluorescence intensity between the supernatant and pellet fractions. The percentage release of SMAC-mCherry was calculated as [Fsupernatant /(Fsupernatant +Fpellet)] x 100.

## In vitro FRET Assays

FRET experiments were carried out as previously described either in the presence or absence of liposomes (*Pogmore et al., 2016*; *Kale et al., 2014*). Briefly, using the same assay buffer that was used to make the liposomes, 100 uL reaction volumes were prepared in a 96-well half-area non-binding surface plate (Corning). Fluorescence intensity of the background was recorded for 30 minutes at 37 °C (*back*). Next, donor labelled protein (e.g., PUMA$^{*A568}$) was added to each well, and the fluorescence intensity measured for 30 minutes at 37 °C ($F_D$). Finally, acceptor labelled protein, (e.g. BCL-X$_L$$^{*A647}$) was added at the desired concentrations to each well, and the fluorescence intensity of the donor was measured for 60 minutes at 37 °C ($F_{DA}$). The FRET efficiency expressed as a percentage was calculated using the following formula:

$$\% \, FRET\, E = \left( 1 - \left( \frac{F_{DA} - back}{F_D \ - back} \right) \right) * 100$$

As a control the fluorescence intensity of the donor ($F_D$) was recorded in the presence of unlabeled acceptor. Therefore, for each concentration of labelled acceptor protein tested ($F_{DA}$), a separate reaction containing the same concentration of unlabeled acceptor protein was measured ($F_D$).

## Cell Lines and Culture

Baby mouse kidney (BMK) cells (wildtype and dko) were cultured in DMEM supplemented with 10% fetal bovine serum (Gibco) and 5% non-essential amino acids (NEAA, ThermoFisher). The BMK-dko cell line was a kind gift from the originator Eileen White, Rutgers Cancer Institute of New Jersey. NMuMG cells (a generous gift of J. Wrana, Lunenfeld-Tanenbaum Research Institute, Toronto, Canada) were cultured in DMEM, containing 10 µg/ml bovine insulin (Sigma), 10% fetal bovine serum (Gibco), and penicillin/streptomycin (Wisent). NMuMG cells expressing fluorescent landmark proteins were generated as previously described (*Schormann et al., 2020*). The originating cell lines used in the studies reported here (BMK-wt, BMK-dko, NMuMG) and all stably transfected clones were shown to be free of mycoplasma using a PCR based test (*Hopert et al., 1993*). All cell lines were maintained in a 5% CO2 atmosphere at 37 °C.

## Immunofluorescence

For immunofluorescence experiments to visualize the localization of endogenous PUMA, MCF-7 cells expressing either mCherry-ActA (mitochondrial marker) or mCherry-CB5 (ER marker) were seeded into 384-well plates, at a density of 7000 cells/well. The next day, cells were treated with indicated drugs and concentrations for 24 hours. On the third day, the cells were stained with DNA stain (DRAQ5) plus MitoTracker Green, or DRAQ5 plus Bodipy-thapsigargin for a total of 30 minutes. Subsequently, all cells were fixed using 4% formaldehyde diluted in Dulbecco's Phosphate Buffered Saline (DPBS). Blocking solution (normal goat serum 5% v/v, Tiriton-X-100 0.3% v/v, diluted in DPBS) was then applied for 1 hour, followed by overnight incubation with primary anti-PUMA antibody. The next day, cells were washed with DPBS, then incubated with secondary anti-rabbit (Alexa Fluor(R) 488) antibody for 2 hours.

## Quantitative Fast FLIM-FRET (qF³)

Detecting protein interactions in live cells was carried out as previously described (*Osterlund et al., 2022*). Briefly, BMK-dko cells stably expressing a mCerulean3-fused anti-apoptotic protein were seeded at 4000 cells/well into a CellCarrier-384 Ultra Microplate (PerkinElmer). 24 hours later, individual wells were transfected with plasmids encoding Venus fused BH3- proteins using TransIT-X2 reagent (Mirus). Non-transfected wells were treated with transfection reagent alone (no DNA added). Media was changed after 3–5 hours. At this time, selected wells were treated with BH3 mimetic or DMSO. Sample plates were incubated 12–18 hours prior to imaging. Immediately before imaging, fluorescence protein standards were added to empty wells. Fluorescein (10 nM in 0.1 M NaOH) and quenched fluorescein (30 µM Fluorescein in 8.3 M NaI and 100 mM $Na_2HPO_4$ (pH 10)) were also added to the plate for instrument calibration. Data acquisition was performed on the INO-FHS microscope as previously described (*Osterlund et al., 2019*).

Data from each replicate was analyzed to generate binned binding curves. All biological replicates (3 or more) were combined and fit to a Hill equation to generate binding curves and calculate dissociation constants. A 1:1 binding between donor (mCerulean3) fused protein and acceptor (Venus) fused protein was assumed for all protein pairs. Regions of interest (ROIs) were automatically identified within images via a watershed algorithm applied to the TCSPC channel. Pixels within each ROI were combined and used to calculate the fluorescence lifetime. Change in angular frequency is then calculated from corresponding phasor analysis as described in *Osterlund et al., 2022*. The intensity of mCerulean3 and Venus per ROI was measured and then converted to concentration values based on measured standard curves prepared using recombinant fluorescent proteins and imaged on the same plate. We applied a narrow physiologically relevant range of mCerulean3 expression (1–3 µM) for our binding curves, and 0–50 µM range for Venus. In the 10–20 µM $Venus_{Free}$ range, the difference in angular frequency between positive control (Venus-BH3) and negative (Venus-BH3-4E mutant) must be greater than 0.05 to be considered as sufficient dynamic range in the assay to reliably detect changes. For each curve, the Bmax (saturation parameter) was estimated from the median of the points in the far right of each binding curve (30–50 µM $Venus_{Free}$). The minimum sRatio, a threshold value representing the curvature of the binding data used to differentiate binding from collisions, was set to 1.5 based on previous analyses and theoretical modelling (*Osterlund et al., 2022*). For each protein pair data were combined from independent replicates (n≥3), and the average $K_d$ reported. In heatmaps, 'binding' was represented in blue and 'no binding' in red. However, the colors are strictly to enable easy visual comparison of the data. In addition, the numerical value of the apparent $K_d$ values are provided in each cell of the heatmaps.

## FLIM–FRET with ER or mitochondrial segmentation

In all other experiments in this paper, to extract FLIM-FRET binding curves ROIs were selected by applying a watershed segmentation algorithm to the TCSPC image of mCerulean3 (e.g. $^C$BCL-XL). Our standard segmentation approach was run on all data as a positive control for detection of binding throughout the cell (see *Figure 4—figure supplement 2* "mC3 total cell"). BCL-XL is found at the ER, mitochondria and in the cytoplasm (*Kaufmann et al., 2003*; *Osterlund et al., 2023*).

The purpose of this experiment was to express mCherry-tagged landmarks for ER or mitochondria in the same cells as PUMA and BCL-XL, then use the expression of the mCherry landmark to define the boundaries of ROI segmentation. This modification in ROI selection allows us to

examine interactions in the ER verses mitochondria. MCF-7 cells stably expressing both [C]BCL-XL and mCherry (red channel)- tagged markers for ER (mCherry-Cb5) or mitochondria (mCherry-ActA) (*Osterlund et al., 2023*) were transiently transfected with constructs to express [V]PUMA, [V]PUMA-4E, [V]PUMAd26 or[V]PUMAd26-4E. Data were collected using two imaging configurations as we recently described (*Osterlund et al., 2023*). The two imaging configurations are rapidly exchanged to acquire signal from mCerulean3 and Venus (fluorescent proteins) by time correlated single photon counting (TCSPC) FLIM, and Venus and mCherry by using a 64-channel hyperspectral detector. Altogether, the Venus protein expression was captured in both configurations and may be used to confirm alignment images acquired in configurations 1 and 2. In contrast to our previous publication (*Osterlund et al., 2022*), we observed low transient expression of Venus. As a result, we modified our analysis pipeline to calculate relative Venus expression as a measure of intensity (arbitrary units) from the TCSPC FLIM image, rather than use the hyperspectral data, which requires background subtraction.

In detail, the intensity of signal from mCherry was determined by summing the hyperspectral counts at wavelengths 592–660 nm. The same parameters were used to segment all images of mCherry expression (Red Channel): background threshold (1000), Laplacian of a Gaussian kernel size (*Delbridge et al., 2016*) with sigma (0.13), structural element size (*Wilfling et al., 2012*), minimum ROI size (*Chi et al., 2020*), erode factor (0). Nevertheless, the resulting ROIs selected by the Watershed Algorithm were distinct (example images given in *Figure 4—figure supplement 2A compare* [Ch]ActA Mitochondria to [Ch]Cb5 ER). As expected, ROIs selected based on the mCherry-Cb5 expression appear more elongated and mesh-like compared to the more punctate ROIs selected based on mCherry-ActA (mitochondria). We provide the MATLAB analysis package on DataVerse (https://doi.org/10.5683/SP3/ZKXQW8).

The ROIs selected from mCherry-marker expression, were directly applied to extract for each ROI the average intensity and lifetime of mCerulean3 (donor) and the average intensity of Venus (acceptor). These data were used to generate FLIM-FRET binding curves in *Figure 4—figure supplement 2B* for ER- versus mitochondria-segmented ROIs as described (*Osterlund et al., 2023*). The % FRET efficiency was calculated using the phasor approach (*Osterlund et al., 2022*; *Ranjit et al., 2018*) and "Acceptor:Donor intensity ratio" represents the ratio of the average intensity per ROI of Venus to mCerulean3 both recorded by TCSPC. During acquisition, a 10 nM fluorescein sample in 0.1 M NaOH was used to standardize the instrument response for the three biological replicates to enable combining the binding curve data.

## Image Colocalization

Analyses of colocalization by image classification were carried out as previously described (*Schormann et al., 2020*). Conventional colocalization studies to measure Pearson's correlation coefficients were carried out in BMK-dko cells. We utilized multiple fluorescent proteins and dyes to validate protein localization. To identify the endoplasmic reticulum, we either used a dye (BODIPY FL thapsigargin, ThermoFisher Scientific) or an overexpressed ER resident protein composed of EGFP fused to the ER specific tail anchor of BIK ([C]BIK). To identify mitochondria, the dyes MitoTracker Red and MitoTracker Green (ThermoFisher Scientific) were used. The cells were also stained with the nuclear stain, DRAQ5 to permit segmentation and quality control (*Oltersdorf et al., 2005*). Query proteins fused to the fluorescent proteins mCerulean3 or Venus were expressed by transfection of BMK-dko cells with gene encoding plasmids. Images were acquired using a 63 X water immersion objective, on the Opera Phenix microscope (PerkinElmer) and colocalization analyzed using an analysis pipeline created in Harmony software. In this pipeline, the DRAQ5 fluorescence signal was used to identify images of individual cells and create a mask for the nuclear and cytoplasmic areas. A low-intensity fluorescence threshold was applied to all cells in order to select for those successfully transfected and expressing the query protein (i.e. Venus or mCerulean3 fused BH3 only protein). Background was subtracted for each image and a Pearson's correlation coefficient between fluorescent protein-fused query protein and fluorescently labelled organelle was calculated for all remaining objects within the cytoplasmic area. The median Pearson's correlation coefficient for each replicate is plotted in Graphpad Prism software with a line indicating the average between all three replicates.

## Cell death assay

Cell death in response to exogenous expression of ᵛBH3-only proteins in HEK293, BMK, and HCT116 cells, was measured as described (*Chi et al., 2020*). Briefly, cells were trypsinized and seeded in CellCarrier-Ultra 384-well plates. One day later, cells were transfected with plasmids encoding ᵛBimL or ᵛPuma (or the mutant proteins) using Mirus TransIT-X2 transfecting reagent. The media was aspirated prior to the addition of the transfection mix into each well. Four hr after transfection, the media was exchanged to remove the transfection mix. To measure cell death in response to BH3 mimetic treatments, cells were seeded as described above and the BH3 mimetics were added the next day as described (*Chi et al., 2020*). After 24 hr of transfection or drug treatment, cells nuclei were stained with Hoechst 33342 and Alexa 647 labelled Annexin V was added to detect externalized phosphatidylserine on the outer leaflet of the plasma membrane in dead or dying cells. Stained cells were imaged by automated confocal microscopy (Opera Phenix, PerkinElmer) and analyzed with Harmony software (V4.9) to obtain %Annexin V positivity for the population of cells in each treatment. Cell death due to ᵛBimL or ᵛPuma expression was quantified as percentage of Venus positive, Annexin V positive cells out of all Venus positive cells. Background Venus intensities in the un-transfected cells were used to determine the threshold for Venus positivity (2 standard deviations above the mean Venus intensity in the un-transfected wells).

## Image-based analysis of subcellular localization by machine learning

NMuMG cells expressing either EGFP-BIK-L61G or EGFP-PUMA4E were seeded (3000 cells per well) in multiple wells (at least 3) of a 384-well microplate (CellCarrier-384 ultra, B128 SRI/160; Perkin Elmer) and allowed to grow for 24 hr before staining with the nuclear dye DRAQ5 (5 nM; Biostatus). Cells were imaged (>15 field of views) on a spinning disk automated confocal microscope (PerkinElmer) with a 40 x water objective (NA = 0.9) in a defined temperature (37 °C) and $CO_2$ (5%) environment. Images were collected using 3-Peltier cooled 12-bit CCD cameras (Type sensiCam, camera resolution 1.3 megapixels; PCO.imaging), unbinned. Segmentation and feature extraction were carried out as described in *Schormann et al., 2020*. Subcellular localization of cell images was determined by using a Random Forests classifier. Landmarks were expressed as EGFP fusion proteins in NMuMG cells for cell organelles shown in *Figure 4*: ER, (Cytochrome b5, BIK) and Calr-KDEL (ER targeting sequence of calreticulin fused to the N-terminus of EGFP and the ER retention sequence KDEL at the C-terminus of EGFP); ERGIC,ERGIC53; Golgi, GalT (N-terminal 81 amino acids of human beta 1,4-galactosyltransferase) and Golgin84; Mitochondria, MAO (sequence encoding the TA region of Monoamine oxidase A) and CCO (Cytochrome c oxidase, subunit VIII); MAM, Phosphatidylserine synthase 1; RAB5A and RAB7A vesicular compartments, Ras-related protein Rab5A and Rab7A; secretory pathway, VAMP2 and VAMP5 (Vesicle-associated membrane protein 2 and Vesicle-associated membrane protein 5); cytoplasm, Vesicle-associated membrane protein 1 A without TMD (delta-TMD-VAMP1); nuclear membrane, Lamin A; lysosomes, LAMP1 and peroxisomes, PTS1 (Peroxisome targeting signal 1).

## Acknowledgements

The authors wish to thank Dr. Justin Kale, Dr. Qian Liu, and Dr. Philipp Mergenthaler for their thoughtful critique and support of this project. This work was supported by Canadian Institutes of Health Research (CIHR) grant FDN143312 (DWA), Canada Foundation for Innovation (DWA), Ontario Ministry of Research and Innovation (DWA), CQDM (DWA), and Canada Research Chairs Program (DWA).

## Additional information

### Funding

| Funder | Grant reference number | Author |
| --- | --- | --- |
| Canadian Institutes of Health Research | FDN143312 | David W Andrews |

| Funder | Grant reference number | Author |
|---|---|---|
| Canada Research Chairs | Tier 1 | David W Andrews |
| Canada Foundation for Innovation | | David W Andrews |
| Ontario Ministry of Research and Innovation | | David W Andrews |
| CQDM | | David W Andrews |

The funders had no role in study design, data collection and interpretation, or the decision to submit the work for publication.

### Author contributions

James M Pemberton, Conceptualization, Formal analysis, Investigation, Visualization, Methodology, Writing – original draft, Writing – review and editing; Dang Nguyen, Elizabeth J Osterlund, Formal analysis, Investigation, Methodology, Writing – review and editing; Wiebke Schormann, Justin P Pogmore, Formal analysis, Investigation, Writing – review and editing; Nehad Hirmiz, Software, Validation, Methodology, Writing – review and editing; Brian Leber, Conceptualization, Project administration, Writing – review and editing; David W Andrews, Conceptualization, Supervision, Funding acquisition, Project administration, Writing – review and editing

### Author ORCIDs

James M Pemberton ⓘD http://orcid.org/0000-0001-8386-1081
Dang Nguyen ⓘD http://orcid.org/0000-0002-3000-2053
Elizabeth J Osterlund ⓘD http://orcid.org/0000-0003-0941-7630
Wiebke Schormann ⓘD http://orcid.org/0000-0003-3055-2706
Justin P Pogmore ⓘD http://orcid.org/0000-0003-4198-2779
David W Andrews ⓘD http://orcid.org/0000-0002-9266-7157

### Decision letter and Author response

Decision letter https://doi.org/10.7554/eLife.88329.sa1
Author response https://doi.org/10.7554/eLife.88329.sa2

## Additional files

### Supplementary files
• MDAR checklist

### Data availability

We provide the MATLAB data analysis package on DataVerse (https://doi.org/10.5683/SP3/ZKXQW8).

The following dataset was generated:

| Author(s) | Year | Dataset title | Dataset URL | Database and Identifier |
|---|---|---|---|---|
| Osterlund E, Hirmiz N | 2023 | Pemberton2023_ERMitoSegmentationcode | https://doi.org/10.5683/SP3/ZKXQW8 | Borealis, 10.5683/SP3/ZKXQW8 |

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
