## [Editor Report]

This is an important study investigating interactions of the pro-apoptotic PUMA with anti-apoptotic BCL-2 proteins. The authors demonstrate convincingly that the PUMA/BCL-2 interactions are mediated not only via BH3-domain interaction, but also depend on a C-terminal sequence of PUMA similar to BIM. Unexpectedly they find PUMA is often localising to the ER. This manuscript is important for researchers focusing on cell death and anti-tumor drug development.

---

## [Decision Letter]

[Editors' note: this paper was reviewed by Review Commons.]

---

## [Author Response]

Reviewer #1 (Evidence, reproducibility and clarity (Required)):In this paper, the authors present convincing experimental proof on why the BH3-only protein PUMA resists displacement by BH3-mimetics, while others such as tBID do not. Using a SMACmCherry based MOMP assay on isolated mitochondria, FRET in the presence of liposomes with a phospholipid composition similar to that of mitochondria as well as quantitative fast fluorescence lifetime imaging microscopy (Frster resonance energy transfer - qF3) they show that the C-terminal region of PUMA (CTS), together with its BH3-domain, effectively "doublebolt" locks its interaction with BCL-XL and BCL-2 to resist displacement by the BCL-XL-specific BH3-mimetic A-1155463 or the BCL-2/BCL-XL inhibitor ABT-263 and AZD-4320. Although a similar mechanism has previously been published for BIM, the novel C-terminal binding sequence in PUMA is unrelated to that in the CTS of BIM and functions independent of PUMA binding to membranes. First, in contrast to BIM, PUMA contains multiple prolines and charged residues, and an unusually short span of hydrophobic amino acids, secondly, full length PUMA was more resistant to BH3-mimetic displacement than a PUMA mutant lacking the CTS (PUMAd26) even in solution suggesting that the CTS of PUMA contributes to BH3-mimetic resistance even in the absence of membranes.The second, quite unexpected finding of this paper is that, in contrast to previous publications, the CTS of PUMA does not target the protein to mitochondria but to the ER. The authors show this by FLIM-FRET imaging and confocal microscopy, and they created mutants to identify the CTS residues (I175 and P180) that mediate binding to ER membranes.The authors did an excellent job to show the mechanism of displacement resistance of PUMA from BCL-2 survival factors from different angles (in vitro, on isolated mitochondria, liposomes and inside living cells), generating respective BH3 and CTS mutants and also domain swaps with other BH3-only proteins such as tBID. Also, the unexpected finding that PUMA primarily localizes to the ER has been extensively scrutinized and the data presented are convincing.

We appreciate the favourable comments and that the reviewer found the data presented convincing.

Major comments:I have only three questions which I like the authors to address before this MS can be published.1) How can PUMA perform its pro-apoptotic action on MOMP from its site on the ER? DoesPUMA eventually localize to MAMs (mitochondrial/ER contact sites)? Is it possible to co-IP PUMA with BCL-XL or BCL-2 from ER membranes or show such an interaction inside cells with PLA?

The reviewer raises an important point. One of the main conclusions from this paper is that the primary localization of exogenously expressed PUMA is at the ER. Our intent was to highlight the inherent specificity of the PUMA CTS sequence. However, we agree that identifying the localization of PUMA-BCL-XL complexes would add significantly to the manuscript. We carefully considered using co-IP or a proximity ligation assay (PLA) in order to investigate the localization of PUMA-BCL-XL complexes. In our experience the use of co-IP is very difficult to interpret due to the well characterized detergent-induced artifacts previously shown for BCL-2 family protein interactions (PMID: 9553144, PMID: 33794146). Moreover, PLAs are a proximity assay with a detection range of ~>20nm, and are difficult to quantify beyond enumerating frequency (ie counting spots). In contrast, the detection of FRET by fluorescence lifetime imaging microscopy (FLIM) is very sensitive to distance with a maximum that is <10nm, and the results can be interpreted quantitatively as apparent dissociation constants (manuscript Figures 2-3). Therefore we elected to use FLIM-FRET to address this question. We examined PUMA-specific interactions with BCL-X_L_ at the ER and mitochondria by differentially segmenting the FLIM-FRET image data based on the signal from a mCherry-fused landmark expressed at the ER (mCherry-Cb5) or mitochondria (mCherry-ActA). This approach has similar spatial resolution to PLA yet retains more rigorous requirement for proximity and the quantitative interpretability of FLIM-FRET.

For these experiments we used a recently described the method of mitochondrial image segmentation using hyperspectral image data collected during FLIM-FRET imaging (Osterlund et al., 2023). In this approach, a watershed segmentation algorithm was used to identify mitochondria areas from mCherry-ActA images collected simultaneously with the FLIM data. The ER was identified in separate samples using the same approach with mCherry-Cb5 image data. Simultaneous collection of the images ensures that the data are not affected by movement within the cells. Example images showing the segmentation results for each organelle have been added to the manuscript as Figure 4 - Figure Supplement 2A.

The results of this FLIM-FRET experiment described in the text lines 581-598, revealed that ^V^PUMA interacts with ^C^BCL-X_L_ within both ER and mitochondria-segmented ROIs (new Figure 4 - Figure Supplement 2B). These results can be explained by the fact that ^V^PUMA is targeted to the ER, and BCL-X_L_ is known to localize to the ER and mitochondria when bound to BH3 proteins in cells (Kale et al., 2018, PMID: 29149100). This result is similar to what we reported for BIK, another ER-localized BH3 protein that exerts its pro-apoptotic function from ER membranes (PMID: 11884414 and PMID: 15809295). Our recent data for ER localized BIK binding to mitochondria-targeted BCL-XL (Osterlund et al., 2023), suggests that, as the referee suggested, binding to occurs via a membrane-spanning interaction at MAMs (ER-mitochondia contact sites) and/or via relocalization of BIK and/or BCL-XL in response to their co-expression (Osterlund et al., 2023). Consistent with these interpretations, when expression of endogenous PUMA was upregulated in response to stress (Figure 4- figure supplement 3A-B), the amount of PUMA increased at both ER and mitochondria (Figure 4- figure supplement 3C). We have presented this data and interpretation on lines 599-621 and discussed the localization results and the similarity to BIK in the manuscript discussion, lines 1029-1035.

2) Since PUMA seems to be "double-bolt" locked to BCL-2 or BCL-XL via its BH3-domain and CTS, how can it act as a pro-apoptotic inducer? Is its main function to act as an inhibitor of BCL-2 and BCL-XL rather than a direct BAX/BAK activator? And if it acts as a BAX/BAK activator, how can it be released from BCL-2/ BCL-XL, for example by another BH3-only protein which is induced by apoptosis stimulation? Or would in this case PUMA remain bound to BCL-2/ BCL-XL in order to activate BAX/BAK (which would be a kind of new activation mechanism)?

We appreciate the reviewers queries and have clarified the text to indicate that our interpretation is that by binding to BCL-XL, PUMA releases active BAX that is sequestered by BCL-XL (as shown in Figure 1A for purified proteins). Double bolt locking increases both affinity and avidity of PUMA for BCL-XL enabling competition to favor PUMA binding and displacement of sequestered BAX. To further address the reviewers point we added two additional experiments now shown in figure supplements to Figure 1. The data shown in new Figure 1 – figure supplement 1A (described on lines 182-191 of the revised manuscript) demonstrates that PUMA kills HCT116 and BMK cells but not HEK293 cells. New Figure 1 – figure supplement 1B shows that inhibition of BCL-XL and MCL-1 using BH3 mimetics is sufficient to kill HCT116 and BMK cells while HEK293 cells are not killed by even high concentrations of these BH3 mimetics. To kill HEK293 cells requires activation of BAX (described on lines 191-201). Together this data indicates that the primary pro-apoptotic function of PUMA is inhibiting BCL-XL and MCL-1 rather than by activating BAX. This data fits very well with PUMA double-bolt locking resulting in very tight binding of PUMA to BCL-XL and likely MCL-1 as the primary mode of PUMA mediated induction of cell death, at least in the three cell lines investigated here. The importance and role of PUMA mediated BAX activation is an interesting area of active investigation that is beyond the purview of the current paper.

3) Is PUMA still bound to the ER when it is transcriptionally induced by genotoxic stress. In this case, the extra amount of PUMA produced is supposed to directly activate BAX/BAK. Does it do this on the ER or on mitochondria?

The referee raises a very interesting point. Interestingly, Zheng et al., 2022 highlighted a P53dependent death response to genotoxic stress, which results in the extension of peripheral, tubular ER and promotes the formation of ER-mitochondria contact sites (PMID: 30030520). Furthermore, PUMA is transcriptionally activated by P53 (PMID: 17360476). Therefore, we hypothesized the induction of PUMA would increase the fraction of PUMA at ER membranes and MAMs. As the latter resemble mitochondria in micrographs of cells we anticipated an increase in apparent mitochondrial localization. To address this question experimentally, we treated MCF-7 cells with genotoxic stress and ER stressors and tracked the expression of endogenous PUMA by immunofluorescence. The results are described in the manuscript (line 603-613, page 28) and shown in Figure 4 figure supplement 3. The immunofluorescence data confirmed that PUMA protein levels increase after genotoxic stress, as expected (Reference 39, 40 in the manuscript) and to a lessor but still significant extent after ER stress (Figure 4 figure supplements 3A and B). In response to stress the amount of PUMA increased at both ER and mitochondria, however, in unstressed cells the endogenous Puma co-localized more to the mitochondria than to the ER (Figure 4- figure supplement 3C). This suggests that similar to BIK localization of PUMA is dynamic. In particular, the abundance and localization of PUMA binding partners such as BCL-X_L_ also affects PUMA localization (the new data are described on pages 2728, Lines 591-621). As described above, the extra PUMA induced by genotoxic stress can indirectly activate BAX by binding BCL-X_L_ and displacing sequestered activated BAX. Our FLIMFRET data suggest PUMA can bind BCL-X_L_ at both the mitochondria and the ER. Moreover, given the expansion of ER-mitochondrial contact sites that occurs during stress we cannot rule out the possibility that ER-localized PUMA can inhibit mitochondria-localized anti-apoptotic proteins (both BCL-X_L_ and MCL-1) at the ER (for BCL-XL) and MAMs for both proteins.

Reviewer #1 (Significance (Required)):Very significant contribution to the field. Quite novelReviewer #2 (Evidence, reproducibility and clarity (Required)):This study by Pemberton and colleagues investigates interactions of pro-apoptotic PUMA with anti-apoptotic BCL-2 proteins, employing a variety of BH3-mimetics. The authors demonstrate that the PUMA/aa BCL-2 interactions are mediated not only via BH3-domain/groove interactions, but also dependent on a C-terminal sequence of PUMA. This mirrors (with distinct differences) what the authors have previously reported for BIM. They then, reveal that unexpectedly PUMA is often localising to the ER (as opposed to mitochondria), though this localisation is not important for the resistance of PUMA/BCL-2 complexes to BH3-mimetic treatment, authors speculate that ER localised PUMA may have a day job.In my opinion, the study is important for several reasons, not least it strongly argues that BH3mimetics are not optimal (in themselves) to promote apoptosis dependent on PUMA, and that approaches to disrupt the "double-lock" mechanisms should be sought - this has clear clinical importance, but equally important is it adds a new layer of complexity to how BCL-2 family members "work", how the double-lock mechanism is overcome in physiological apoptosis remains an open question, for instance. The data support the authors' conclusions, I have a few points that could be addressed.

The positive comments from the reviewer are greatly appreciated.

1. The authors data in cells is consistent with a membrane recruitment effect of the PUMA CTS making a contribution to the resistance of PUMA/aa BCL-2 complexes to BH3-mimetics. What I found really intriguing, is that the CTS also influences affinity in the absence of membranes (Figure 1) - could the authors speculate why they think CTS may be affecting PUMA/aaBCL-2 binding in the absence of membranes?

We agree with the reviewer that membrane binding contributes to BH3 mimetic resistant binding of PUMA to BCL-XL consistent with elegant data presented previously (Pécot et al., 2016; PMID: 28009301). However, we show in Figure 5D that mutants of ^V^PUMA-d26 with restored membrane binding (^V^PUMA-d26-ER1 and ^V^PUMA-d26-ER2) remain sensitive to BH3-mimetic displacement, indicating that membrane binding alone is not sufficient to confer resistance to BH3-mimetics. Furthermore, as the reviewer pointed out BH3 mimetic resistant binding is observed in the absence of membranes (Figure 1).

The data using purified proteins strongly suggests that the CTS of PUMA binds to BCL-XL and is directly involved in the protein-protein interaction. The fact that PUMA with the C-terminal fusion to the fluorescent protein Venus (PUMA^V^) still localizes to membranes in live cells (Figure 4 D,E) suggests that the C-terminus of PUMA does not span the membrane bilayer. Instead, we hypothesize that the C-terminus of PUMA binds peripherally to the membrane making it available to physically contribute to a protein interaction with anti-apoptotic proteins. This interpretation is consistent with the low hydrophobicity and high proline content (6 of 28 residues) of the amino acid sequence of the PUMA CTS as shown in Figure 6 and compared to the transmembrane tail anchor sequences of other proteins, including the BH3-protein BIK, in Figure 5 supplement 1. Binding of Bcl-XL by both the BH3 region and CTS of PUMA would increase both the affinity and avidity of the interaction. The presentation of this data has been revised to add clarity on pages Page 8, lines 215-223 and in the discussion (Lines 988-997 and 1044-1050).

2. A minor point for clarification, are the mitochondria used in Fig 1A from BAX/BAK DKO cells?- I had presumed so given exogenous BAX was added, but didn't note this in the text.

We indeed use mitochondria from BAX/BAK DKO cells and exogenous recombinant BAX in Figure 1A. This has now been added to the text on lines 166-180.

Reviewer #2 (Significance (Required)):Detailed in report above.Reviewer #3 (Evidence, reproducibility and clarity (Required)):In this paper, Pemberton et al show that PUMA resists BH3-mimetic mediated displacement from BCL-XL via a novel binding site within its C-terminus of PUMA termed CTS (the last 26aa). Interestingly, the CTS of PUMA directs the protein to the ER membrane and residues I175 and P180 within the CTS are required for both ER localization and BH3-mimetic resistance.Specific comments:1. BH3-mimetics kill cells by displacing sequestered pro-apoptotic proteins to initiate apoptosis.However, PUMA resists BH3-mimetic mediated displacement, and PUMA-d26 and PUMAI175A/P180A (CTS) do not. Thus, are these mutants sensitive to BH3-mimetics cell killing? In other words, do BH3-mimetics kill PUMA-/- cells that express either PUMA-d26 or PUMA I175A/P180A but not PUMA-/- cells that express wild type PUMA?

The reviewer raises a very interesting question that unfortunately we have been unable to address unambiguously. To answer this question requires separating the effects of PUMA on anti-apoptosis proteins and on activation of BAX and BAK as exogenous expression of either PUMA-d26 or PUMA I175A/P180A is sufficient to kill PUMA-/- cells without the addition of a BH3 mimetic. To date we have been unable to identify mutants that inhibit anti-apoptotic proteins but that do not activate BAX and BAK as both PUMA-d26 and PUMA I175A/P180A have impaired BAX-activation function. This is additionally complicated by PUMA mediated inhibition of MCL-1, BCL-2 and BCL-W. Further, it isn’t possible to separate the function(s) using BAX/BAK knock-out cells because then PUMA induced cell death is completely abrogated. Understanding the direct activation of BAX by PUMA is an area of current investigation that is out of the scope of this paper as here we are focused on the interaction(s) of PUMA with anti-apoptotic proteins.

2. The authors elegantly demonstrate using microscopic analysis that over expressed PUMA mostly localizes to the ER membrane. Since this is a major conclusion in the paper which is different than previously reported, the authors should confirm these findings using sub-cellular fractions followed by Western blot analysis. They should demonstrate that endogenous and over-expressed PUMA are mainly localized to the ER membrane and that the PUMA-d26 and PUMA I175A/P180A are mainly localized to the cytoplasm.

We appreciate that the reviewer found the microscopic analysis convincing. We also tested the idea of sub-cellular fractionation proposed by the reviewer. However, we have found it to be very difficult to separate mitochondria and MAMs. To address the question raised we instead performed new co-localization experiments, in addition to those reported for PUMAd26 and the point mutants in Figure 6 (images in Figure 6 - figure supplement 3). The new experiments are for endogenous PUMA at steady state and with increased expressed in response to stress. These immunofluorescence experiments are reported in Figure 4 -figure supplements 3. We also added FLIM-FRET experiments in which ROIs were derived from areas of the cell enriched in either ER or mitochondria (Figure 4 - figure supplement 2). The results of these experiments indicate that PUMA localization is dynamic and are described in detail above in response to reviewer 1 question 3 and in the manuscript from line 579 to 621 and discussed on lines 1029-1036.

Reviewer #3 (Significance (Required)):The advance in this paper is significant and the paper should be published once the specific comments are adequately addressed.